# Electron-mediated control of nanoporosity for targeted molecular separation in carbon membranes

Banseok Oh[1], Hyeokjun Seo[1], Jihoon Choi[1], Sunggyu Lee[1] & Dong-Yeun Koh [1,2] ✉

Carbon molecular sieve (CMS) membranes are considered game-changers to overcome the challenges that conventional polymeric membranes face. However, CMS membranes also confront a challenge in successfully separating extremely similar-sized molecules. In this article, high-precision tuning of the microstructure of CMS membranes is proposed by controlled electron irradiation for the separation of molecules with size differences less than 0.05 nm. Fitting CMS membranes for targeted molecular separation can be accomplished by irradiation dosage control, resulting in highly-efficient $C_2H_4/C_2H_6$ separation for low dosages (~250kGy, with selectivity ~14) and ultra-selective $H_2/CO_2$ separation for high dosages (1000~2000kGy with selectivity ~80). The electron irradiated CMS also exhibits highly stabilized permeability and selectivity for long-term operation than the pristine CMS, which suffers from significant performance degradation due to physical aging. This study successfully demonstrates electron irradiation as a possible way to construct "designer" nanoporous carbon membranes out of the standard components mostly confined to pyrolysis conditions.

Energy-efficiency improvements in the industrial separation sectors will accelerate the global decarbonization efforts, as these processes account for more than 10% of global energy consumption and carbon emissions[1–3]. For example, the current industrial separation of ethylene from ethane performed with energy-intensive cryogenic distillation solely consumes 0.1% of global energy use due to their extremely similar size and chemical properties[4]. This challenging separation could benefit from the low-energy approach of membranes. Nanoporous carbon membranes (i.e., carbon molecular sieve, CMS), which are derived from the thermal decomposition of polymeric materials, have demonstrated the fine separation of molecules with sub-0.1 nm size differences: gas separations for light hydrocarbons ($C_2$ or $C_3$ compounds), natural gases ($N_2/CH_4$), and liquid hydrocarbon separations for xylene isomers or hexane isomers[5–10]. The CMS provides slit-like transport pathways for the molecules by its disordered $sp^2$-hybridized graphenic layers. The unique bimodal pores of the CMS are composed of supermicropores (0.7–2 nm) and ultramicropores

(<0.7 nm)[11]. The connectivity between these two types of pores in the CMS allows high permeability, which is derived from the pore gallery in supermicropores, and high selectivity, which is due to the limiting pore diameter with sufficient structural rigidity to enhance the diffusion selectivities[12–14].

However, CMS structures have limited separation performance for gas pairs with extremely small size differences (i.e., $\Delta$size < 0.05nm; for example, $C_2H_4/C_2H_6$ or $H_2/CO_2$). This is primarily due to insufficient pore confinement and limited populations of sub-0.5 nm pores. As one example, CMS membranes prepared from Matrimid® and 6FDA(4,4'-(hexafluoroisopropylidene)diphthalic anhydride)-based polyimide were tested for $C_2H_4/C_2H_6$ separation ($\Delta$size = 0.01nm) with different heating protocols during the fabrication of CMS membranes[15]. Although those CMS membranes exhibited moderate $C_2H_4$ permeability (~100 Barrer) in a relatively low pyrolysis temperature range of 500–550 °C, the $C_2H_4/C_2H_6$ selectivity remained at around 3–5, indicating the minor possibility of sharp $C_2H_6$ differentiation by the CMS

[1]Department of Chemical and Biomolecular Engineering (BK four), Korea Advanced Institute of Science and Technology, Daejeon 34141, Republic of Korea.
[2]KAIST Institute for NanoCentury, Daejeon 34141, Republic of Korea. ✉e-mail: dongyeunkoh@kaist.ac.kr

window size. When the pyrolysis temperature was raised to 800 °C, the selectivity was improved to 9 while the $C_2H_4$ permeability dropped an order of magnitude. As another challenging example, $H_2/CO_2$ (Δsize = 0.041nm) selectivity of less than 10 is usually reported in CMS membranes regardless of the final pyrolysis temperature or the types of precursors[16–18]. In addition to the strong adsorption between $CO_2$ and carbon surfaces, which enhances intrinsic $CO_2$ permeability, the incomplete development of sub-0.5 nm ultramicropores in the CMS matrix suppresses the effective discrimination between $H_2$ and $CO_2$.

Several post-synthetic ultramicropore tuning mechanisms for CMS to align the specific pore size have been recently proposed. Hyperaging with mild post-pyrolysis heat treatment allows pore-size tuning to enhance the selectivity of $H_2$ vs. larger hydrocarbon (Δsize ≥ 0.1nm) gas pairs—which pushes pore size distribution of the pristine CMS to a smaller ultramicropore regime (<0.5 nm)[19]. On the other hand, when a trace amount of $O_2$ is present in an inert gas environment during pyrolysis (known as "$O_2$-doping")[20], chemically adsorbed $O_2$ molecules on the carbon edge sites induce the formation of a closed ultramicropore window and result in enhanced permeation performance such as in $CO_2/CH_4$. Likewise, low concentration hydrogen dosage during the pyrolysis created mid-sized ultramicropores (>0.5 nm), which are suitable for separating vapor-phase xylene isomers (i.e., *p-xylene and o-xylene*, Δsize = 0.1nm)[21]. In addition, cellulose hollow fiber membranes which had not been reported before were fabricated and pyrolyzed into CMS membranes without severe structural collapse[22]. The cellulose-originated CMS membranes achieved distinguished $H_2/CO_2$ separating capability (e.g., 101 Barrer of $H_2$ permeability with $H_2/CO_2$ selectivity ~28), indicating the importance of suitable polymeric precursor selection and pyrolysis temperature.

However, the major handles for tuning ultramicropores so far are limited to optimizing the pyrolysis temperature, pyrolysis environment, and precursors. Therefore, it is crucial to seek advanced and adjustable tuning factors for the fine calibration of both ultramicropores and supermicropores of the CMS microstructure to achieve molecular separations for challenging gas pairs with Δsize < 0.05nm.

We present highly-controlled electron beam irradiation as an effective method to precisely tailor the ultramicropore dimensions in CMS membranes (Fig. 1a). It is well known that high-energy electron irradiation causes structural changes in carbon-based materials, including carbon nanotubes (CNTs) and fullerenes ($C_{60}$)[23,24]. Such high-energy interaction can induce point defects in carbon sheets and strands, making vacancies and interstitial junctions on $sp^2$-structures, thereby creating unique carbon landscapes[25–27]. We successfully tuned ultramicropore dimensions by interacting carbon strands with different electron dosages (50–2000 kGy, 1 kGy is equivalent to 1 kJ/kg of energy transfer). We found that electron irradiation drives mid-size ultramicropores to be concentrated below 0.4 nm, which intensifies the size-selective permeation of similarly-sized molecules, such as $C_2H_4/C_2H_6$ and $H_2/CO_2$, which are key energy-intensive separation model pairs. Notably, the permselectivity of the membranes sharply increased 3 to 4-fold when the controlled amount of electrons was irradiated on the carbon membrane, crossing the upper bound limits. We show that both pore tightening and oxygen-rich surface formation by electron irradiation were responsible for performance enhancements. High-energy interaction between electrons and carbon surfaces further allowed the retention of original membrane performance over time with greatly improved aging resistance.

## Results
### Electron beam induced CMS tuning
To investigate the effect of electron irradiation on a model CMS membrane, we selected 6FDA:BPDA(1:1)-DAM [4,4'-(hexafluoroisopropylidene)diphthalic anhydride (6FDA), 4,4'-biphthalic anhydride (BPDA), 2,4,6-trimethyl-1,3-phenylenediamine (DAM), 1:1 indicates equimolar copolymer] as a precursor (Supplementary Fig. 1).

6FDA-based polyimides have shown high gas separation performances and resistance to swelling due to structural rigidity and inefficient chain packing from trifluoromethyl group (-$CF_3$) on the backbone[28,29]. Analogous to the precursor properties, the inefficient packing and high free volume of 6FDA-based polyimide induced an open, selective, and highly porous CMS membrane structure in the pyrolysis process. Therefore, Carbon molecular sieve membranes fabricated with 6FDA-based polyimides indicated superior permeability for gas separation, especially in light hydrocarbons, than other commercialized or studied precursor derived CMS[30–33]. Given the coexistence of bulky 6FDA and planar BPDA monomers, 6FDA:BPDA(1:1)-DAM derived CMS provides a more selective pore with moderate permeability than other 6FDA-based polyimide such as 6FDA-DAM[18,34]. CMS membranes were fabricated with thickness of 75–80 μm by pyrolyzing the symmetric dense film of 6FDA:BPDA-DAM under an ultra-high-purity (UHP) argon environment (Fig. 1a). The CMS were prepared as self-standing and symmetric films then irradiated with a range of 50–250 kGy by a high-energy electron-beam accelerator in ambient air. From our extensive experience with electron-irradiated CMS membranes, we believe that the macro-mechanical properties were not altered upon electron irradiation of CMS membranes. For example, mechanical property test on a bundle of carbon fibers (not membranes, diameter <10 μm) revealed stable tensile stress and elongation after electron irradiation, therefore, mechanical property of electron-irradiated CMS would be insignificantly changed[35,36]. While scanning electron microscope (SEM) cross-section images did not catch apparent differences between the pristine and electron irradiated CMS membranes, energy-dispersive X-ray spectrometer (EDS) scan revealed an asymmetric chemical composition with an oxygen-rich surface of the CMS film after electron irradiation (Fig. 1b-d and Supplementary Fig. 2). High-resolution transmission electron microscopy (HR-TEM) images of the CMS films illustrated the maze-like microstructure of amorphous carbon membranes (Fig. 1e, g). The corresponding Fast-Fourier Transformed (FFT) patterns (Fig. 1f, h) show continuous ring shapes distinguished by vivid contrast differences in the TEM images, indicating the representative pattern of disordered carbon strands. When the pristine CMS was exposed to 250 kGy (250kGy-CMS, hereafter) of electron irradiation, the radial intensity of the FFT patterns changed with different the number of rings, indicating the different packing degrees of the carbon strands. The breadth of radial intensity for 250kGy-CMS was from 5.65 to 3.27 $nm^{-1}$ (corresponding to 0.177 to 0.306 nm of micropore spacing, respectively, Fig. 1h) showing that denser, compact ultramicropore dimensions were obtained when compared to those of pristine CMS from 5.23 to 2.72 $nm^{-1}$ (0.191 to 0.367 nm of micropore spacing, Fig. 1f). Furthermore, the disappearance of radial intensity at 3.94 $nm^{-1}$ (0.254 nm of micropore spacing, Fig. 1f) of the pristine CMS after electron irradiation indicates the formation of condensed ultramicropore dimensions. The formation of tighter ultramicropores reduces the number of accessible transport channels for large gas molecules, thus facilitating the selective discrimination of the closely-sized gas pairs.

To precisely analyze the orientation of the amorphous carbon strands in the electron irradiated CMS, synchrotron grazing-incidence wide-angle X-ray scattering (synchrotron GI-WAXS) was used. GI-WAXS utilizes a very small incidence angle to confine the penetration depth to the nanometer level, probing the molecular length-scale structures in the amorphous CMS membranes[37,38]. 2D scattering images show the ring patterns of both the precursor and the irradiated CMS films (Fig. 2a, b and Supplementary Fig. 3). The anisotropic packing of the CMS structures is observed by the more dominant intensity development of $q_z$ than that of $q_{xy}$, and this tendency becomes more pronounced as the pyrolysis temperature increases (500 °C to 600 °C, Fig. 2b and Supplementary Fig. 3). The '*out-of-plane*' ($q_z$, $q_{out}$) and '*in-plane*' ($q_{xy}$, $q_{in}$) scattering vectors indicate that the aromatic strands constituting the CMS micropores are anisotropic

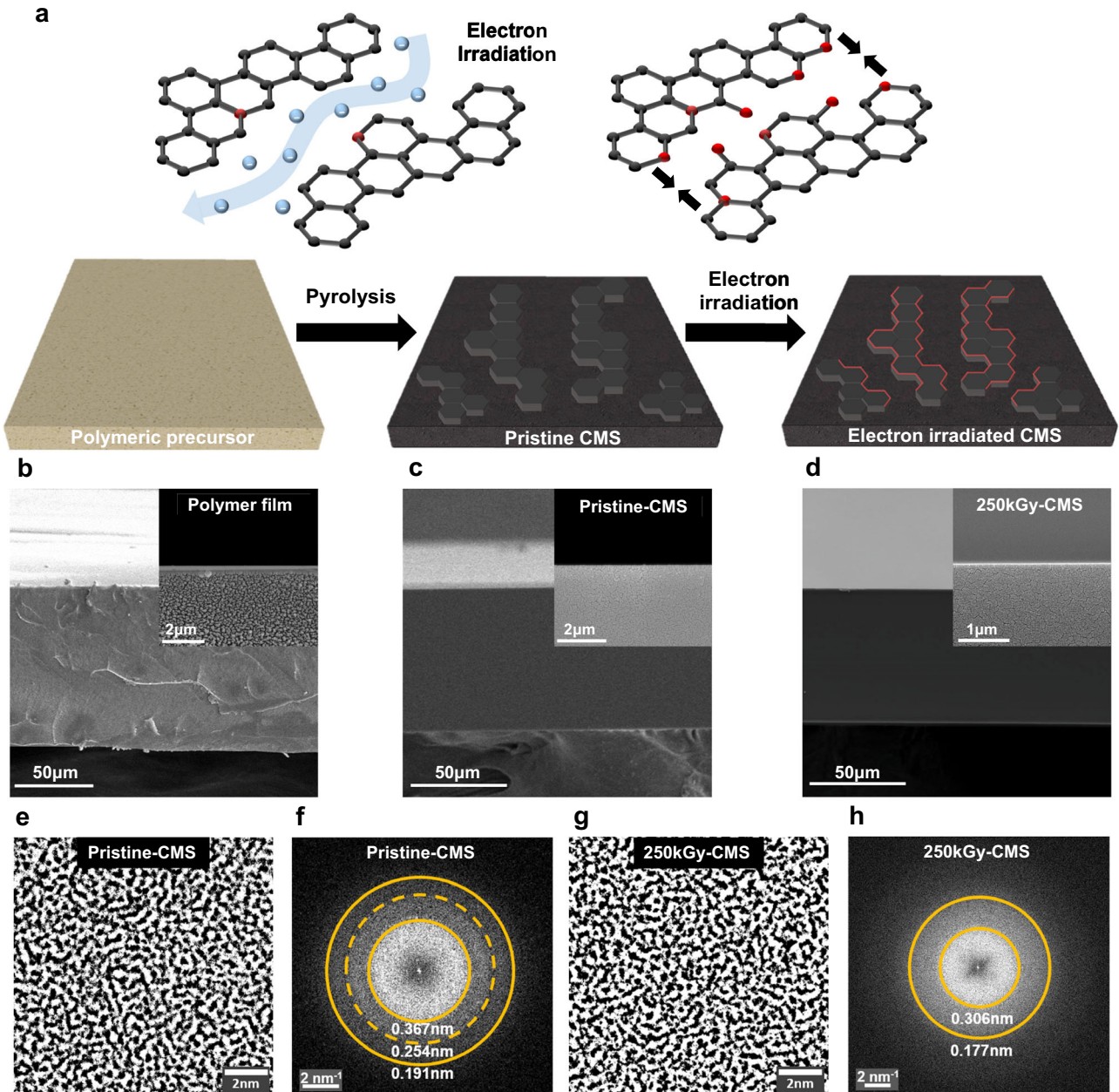

**Fig. 1 | Preparation of electron-irradiated CMS membranes and morphology characterization. a** Schematic description of polymeric precursor pyrolysis for fabricating CMS membranes with electron irradiation with microstructure tuning. **b**–**d** Scanning electron microscopy (SEM) cross-section images and their high magnitude images (inset). **b** Polymeric precursor (**c**) 500 °C pristine-CMS (**d**) 500 °C 250kGy-CMS. **e**, **g** High-resolution transmission electron microscopy (HR-TEM) images of the 500 °C pristine- and 250kGy-CMS with (**f**, **h**) their corresponding 2D Fast Fourier Transform pattern.

with face-on orientation, not edge-on packing perpendicular to the substrate. Notably, 2D scattering image of 250kGy-CMS shows that the "*out-of-plane*" vector ($q_z$) intensity in the range of 0.5 to 1.0 Å⁻¹ was mitigated, while the peak around 1.4 Å⁻¹ became strong and narrow (Fig. 2b, c). The vertical line-cut as a function of $q_z$ shows a broad peak along $q_{out}$ ~1.365 Å⁻¹ involved with imperfect lamellar-like ultra-micropores with approximately 0.46 nm *d*-spacing for 250kGy-CMS and 0.514 nm *d*-spacing for the pristine CMS (Fig. 2c). In addition, the GI-WAXS peak analysis of 250kGy-CMS (Fig. 2d) revealed that the assumed pore size distribution started from 1.635 Å⁻¹ (i.e., 0.385 nm), thus confirming that the sharp molecular sieving of the ethane (0.4 nm) is possible.

We further found that the electron irradiation not only success-fully tuned the ultramicropore dimensions but also modified the sur-face of the CMS via forming oxygen-rich carbon strands. The survey scan of X-ray photoelectron spectroscopy (XPS) compares the atomic composition of the surface on the pristine CMS and 250kGy-CMS, showing the intensified O1s peak (~530 eV) and relatively decreased C1s peak (~285 eV) after electron irradiation (Fig. 2e and Supplemen-tary Fig. 4). It is well known that the high-energy electron interaction generates atomic vacancies, typical carbon defects that are vulnerable to oxidation or reaction with other gas molecules[25,39]. Furthermore, the formation of interstitial defects is responsible for locally reduced carbon dimensions, liable to rearrangement of bonds and topological transformation of the carbon structures. The defective strands exposed to ambient air would combine with the $O_2$ and develop an oxidized surface, creating a narrow ultramicropore window on the CMS surface that can reject large penetrants. However, electron irra-diation can be distinguished from $O_2$ doping treatment on CMS membranes. The $O_2$ doping process uses trace amounts of oxygen

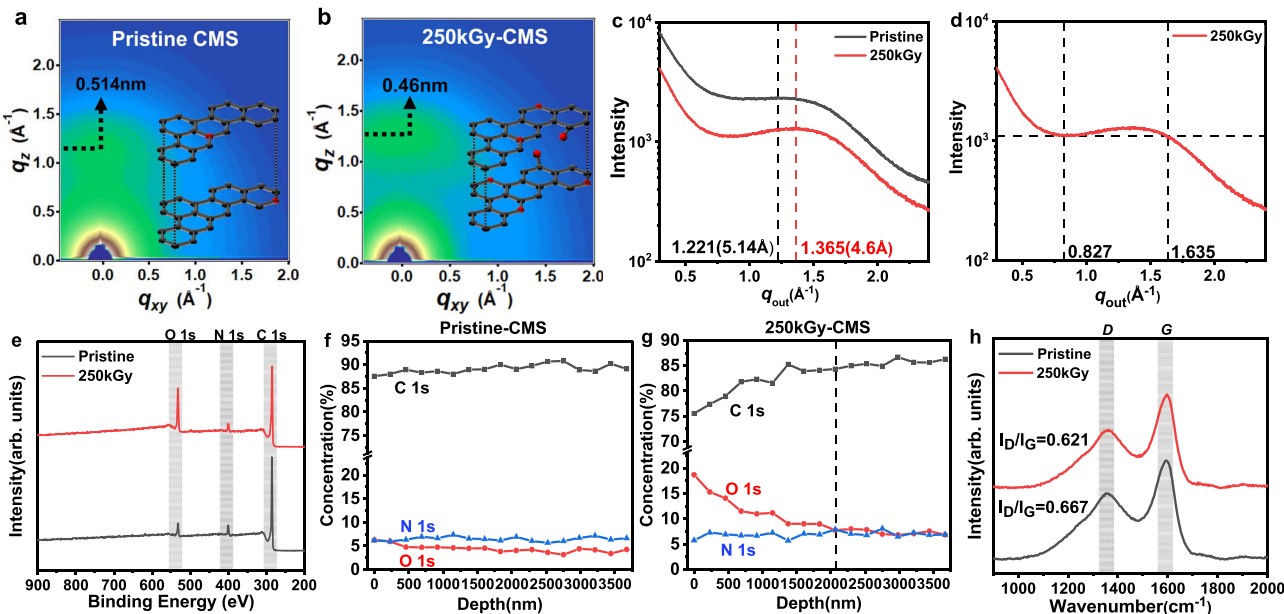

**Fig. 2 | Structure characterization of the pristine and electron-irradiated CMS films. a, b** 2D-Grazing incidence wide angle X-ray scattering (GI-WAXS) patterns of 500 °C pristine- and 250kGy-CMS. **c** 1D-GI-WAXS spectra of the pristine- and 250kGy-CMS along the out-of-plane axis. **d** 1D-GI-WAXS spectra of the 250kGy-CMS with a broad peak highlighted by a dashed line. **e** X-ray photoelectron spectroscopy (XPS) survey scan of the 500 °C pristine- and 250kGy-CMS. **f, g** XPS depth profiles of the 500 °C pristine- and 250kGy-CMS to C, N and O composition. **h** Raman spectrum of the 500 °C pristine- and 250kGy-CMS with $I_D/I_G$ ratios. Source data are provided as a Source Data file.

(<50 ppm) in a pyrolysis environment, and then $O_2$ is selectively chemisorbed onto the highly reactive carbon edge sites. For example, previously reported CMS membranes produced from 6FDA:BPDA(1:1)-DAM showed $C_2H_4$ permeability of 110 Barrer with $C_2H_4/C_2H_6$ selectivity of 4 when pyrolyzed at 550 °C in a UHP Ar environment (<1 ppm $O_2$)[15]. When the $O_2$ concentration in the pyrolysis chamber was increased to 30 and 50 ppm, the CMS membranes showed $C_2H_4$ permeability of 85 and 68 Barrer, respectively, with slight improvements in the $C_2H_4/C_2H_6$ selectivities (5.2–5.5, respectively). While the high-energy electron irradiation directly combines the advantages of a defect-mediated oxidized carbon surface with reduced ultramicropore dimensions, a higher degree of molecular specificity of the resultant membrane is expected. As shown in the depth profiles of XPS (Fig. 2f, g and Supplementary Fig. 5), the pristine CMS shows a consistent atomic composition along with an etched depth down to 3.5 μm from the surface. However, 50 kGy-CMS has an oxidized surface and the oxygen composition continuously decreases with the etched depth. In addition, all the 100kGy-CMS, 250kGy-CMS and 500kGy-CMS show a 3 or 4-fold increase in oxygen composition on the surface, indicating the formation of oxygen-rich carbon surfaces. When the electron irradiation increased from 50kGy to 500kGy, the oxygen composition and the thickness of the oxidized carbon layer also increased (from 1.61 to 4.13 μm Supplementary Fig. 6). Raman spectroscopy was also used to measure the defectiveness of the carbon structure after electron irradiation (Fig. 2h and Supplementary Fig. 7). $D$ peak ($I_D$ ~1350 cm⁻¹) development implies defects in crystalline carbon, but in the case of the disordered carbon networks in the CMS membrane, the $D$ peak represents the proportion of the aromatic rings in the carbon clusters[40]. $G$ peak ($I_G$ ~1580 cm⁻¹) involves in-plane bond-stretching of $sp^2$-carbon atoms and occurs at all $sp^2$ sites, including aromatic and olefinic carbons. Therefore, the defectiveness in carbon materials tends to be the opposite of the $I_D/I_G$ ratio. Since the $I_D/I_G$ ratio of the 250kGy-CMS ($I_D/I_G$ = 0.621) is lower than that of the pristine CMS ($I_D/I_G$ = 0.667), the electron irradiated CMS is assumed to be more defective overall. From this, we can conclude electron irradiation induces not only oxidization on the CMS surface but also defective carbon network within CMS.Thus, we precisely measured the

permeation of gas molecules in these CMS membranes to understand the structure-property relationship in the electron-irradiated CMS membranes.

**Gas permeation and diffusion.** The electron-induced ultramicropore tuning can be directly verified through gas permeability measurements for gas pairs. We first measured the unary gas permeabilities of pristine CMS (pyrolyzed at 500 °C) film and electron-irradiated CMS films (50kGy-CMS, 100kGy-CMS, and 250kGy-CMS) via a constant-volume gas permeation apparatus (Fig. 3b and Supplementary Figs. 8 and 9). When a low electron dosage (50 kGy) was applied to the CMS films, the unary gas permeabilities of relatively small gases(from He to $C_2H_6$) did not dramatically change (Fig. 3b) but sharp molecular cut-off appeared between $C_3$ hydrocarbon gases (i.e. $C_3H_6/C_3H_8$ selectivity increased from 13.28 to 47.13). When the electron dosage exceeded 100kGy, the permeability of the larger gases significantly decreased (e.g., 55% reduction for $CH_4$ and 84% reduction for $C_2H_6$) compared to smaller gases—indicating the formation of tight but size-selective ultramicropores. In addition, the highly irradiated CMS (250kGy-CMS) showed dramatically reduced permeability for most gases (e.g., 95 and 98% reduction for $CH_4$ and $C_2H_6$ respectively, even 90% reduction in $N_2$) except for helium, which means that the ultramicropore distribution of the electron-irradiated CMS had shifted toward the sub-0.4 nm region, with the highest pore populations concentrated at 0.33 to 0.40 nm (Fig. 3a and Supplementary Fig. 10). Especially in the case of $C_2$ hydrocarbons ($C_2H_4/C_2H_6$), the permeability of the ethane drastically decreased to 19.9 Barrer compared to that of ethylene (180.6 Barrer) in 100kGy-CMS, with a significantly increased ideal $C_2H_4/C_2H_6$ selectivity of 9.06 (Supplementary Fig. 11). However, when a very high electron dosage was applied (500 kGy and 1000 kGy), the formation of the tightest structures with negligible ethylene permeabilities were observed (Supplementary Fig. 12).

We compared the single-component separation performance of the electron-irradiated CMS with other state-of-the-art polymeric membranes, CMS membranes, mixed matrix membranes (MMMs), and metal-organic framework (MOF) membranes (Fig. 3c and Supplementary Tables 1–4), and found that separation performances of the

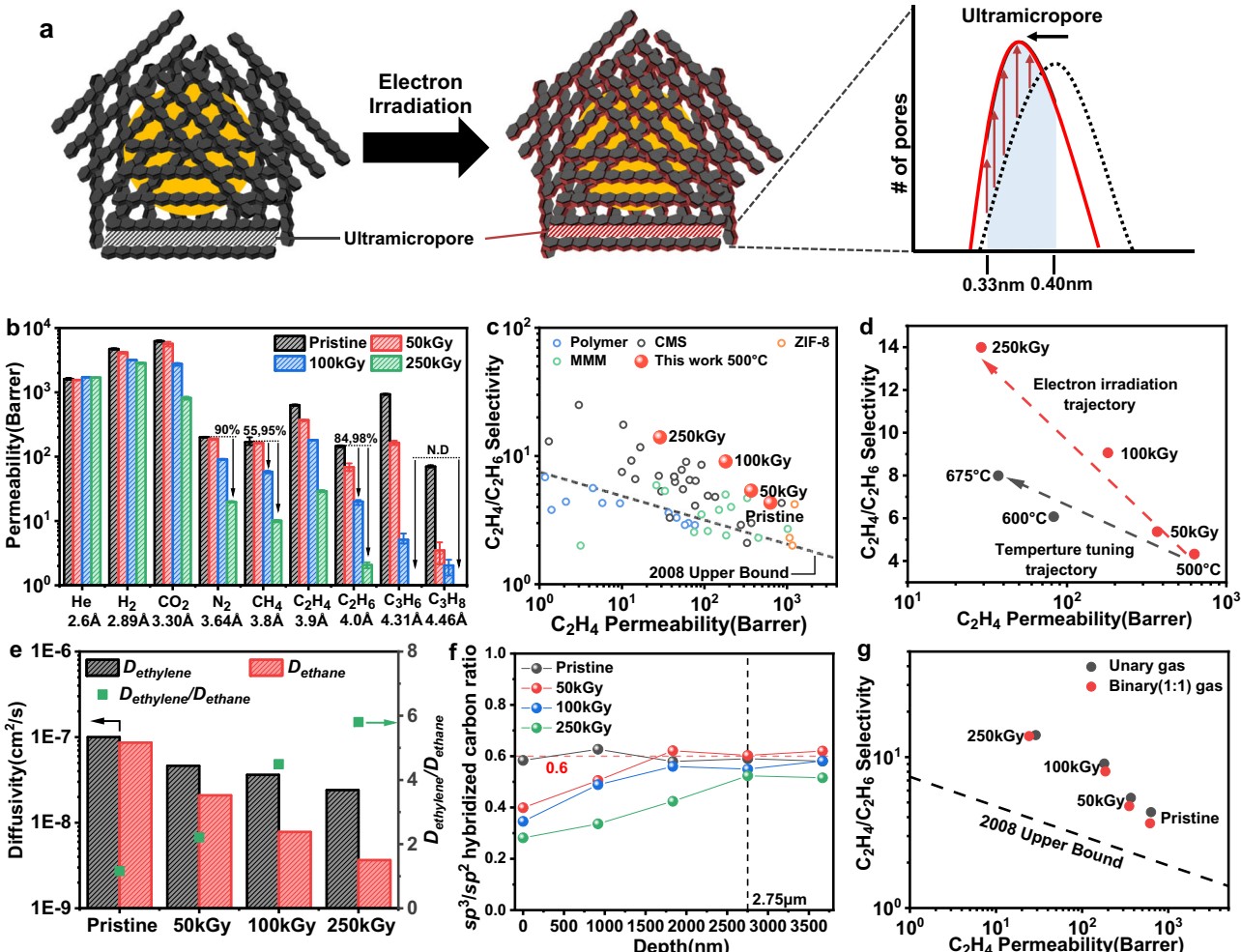

**Fig. 3 | Gas permeation performances and diffusion properties of the prepared CMS membranes. a** Schematic description of turbostratic carbon structure with hypothetical ultramicropore size distribution change before and after electron irradiation. The red-shaded edge of the carbon strands imply oxygen-functionalization and pore compaction. **b** Permeation data of 500 °C pristine and electron-irradiated CMS membranes for nine gases with respect to kinetic diameter. The black arrows show permeability reduction in electron-irradiated CMS (100 and 250kGy-CMS) compared to the pristine one. C₃ gas permeabilities in 250kGy-CMS were not determined due to measurement limit. (error bars indicate the standard deviation at three or four measurements) **c** C₂H₄/C₂H₆

separation performance comparison for 500 °C electron-irradiated CMS membranes and other state-of-the-art membranes. ZIF-8 and MMM represent Zeolitic imidazolate framework-8 and mixed matrix membranes, respectively. **d** Performance investigation of pyrolysis temperature (500-675 °C) and electron irradiation (50-250 kGy). **e** C₂H₄ and C₂H₆ diffusivities of 500 °C pristine- and electron-irradiated CMS membranes and corresponding diffusion selectivity for C₂H₄/C₂H₆. **f** sp³/sp² hybridized carbon ratio of pristine and electron-irradiated CMS (100 and 250kGy) with respect to etched depth. **g** C₂H₄/C₂H₆ permeation data of unary and binary (equimolar mixture) systems. Source data are provided as a Source Data file.

electron-irradiated CMS membranes outperform those of the mentioned membranes. The electron-irradiated CMSs had high permeability and selectivity, for example, the 100kGy-CMS exhibited high C₂H₄ permeability (>200 Barrer) with C₂H₄/C₂H₆ selectivity ~10. Similar ultramicropore tuning was observed when a different pyrolysis temperature (600 °C) was used (Supplementary Fig. 13). The C₂H₄ permeability of 600 °C pristine CMS was 93.1 Barrer and moved along 48.1 and 29.8 Barrer (for 50kGy- and 100kGy-CMS) with a corresponding selectivity increase of 6.0 to 9.7. However, the separation performance of the 600 °C pyrolyzed 250kGy-CMS was not determined due to the measurement limit of very low C₂H₆ permeability (C₂H₄ permeability was calculated to be 10.42 Barrer). To separately investigate the impact of pyrolysis temperature and electron irradiation, three differently pyrolyzed pristine CMS (500, 600, and 675 °C) and one electron-irradiated CMS (500 °C) were compared (Fig. 3d). Beginning with 500 °C pristine CMS (C₂H₄ permeability 630.4 Barrer and selectivity 4.3), changing the pyrolysis temperature drove the CMS structure to be more closed and more selective. For example, when the

pyrolysis temperature was increased to 675 °C, the pristine CMS membrane showed largely decreased C₂H₄ permeability of 37 Barrer with a slight increase in the C₂H₄/C₂H₆ selectivity of 7.9 (following the black colored trajectory in Fig. 3d). Electron irradiation (up to 250kGy) led to similar C₂H₄ permeabilities, but with much-improved C₂H₄/C₂H₆ selectivity up to 14 (following the red colored trajectory in Fig. 3d).

To elucidate the possible reason for the excellent C₂H₄/C₂H₆ separation via electron-irradiation, diffusion analysis was conducted via a custom-built pressure-decay sorption setup (Supplementary Fig. 14). C₂H₄ and C₂H₆ uptake curves for CMS films at different fugacities were fitted by the Fickian diffusion model to calculate their diffusivities. For all CMSs tested, there was a relatively delayed uptake of C₂H₆ over C₂H₄ (Supplementary Figs. 15–17), and the gaps between relative half-time diffusivities (C₂H₄/C₂H₆) increased as the electron dosage increased. Figure 3e shows the Fickian diffusivities of C₂H₄ and C₂H₆ for all CMS films revealing the enhanced diffusion selectivity of C₂H₄/C₂H₆ at higher electron dosages (up to 250 kGy). While the C₂H₄ and C₂H₆ sorption isotherms at 308.15 K (Supplementary Fig. 18) imply

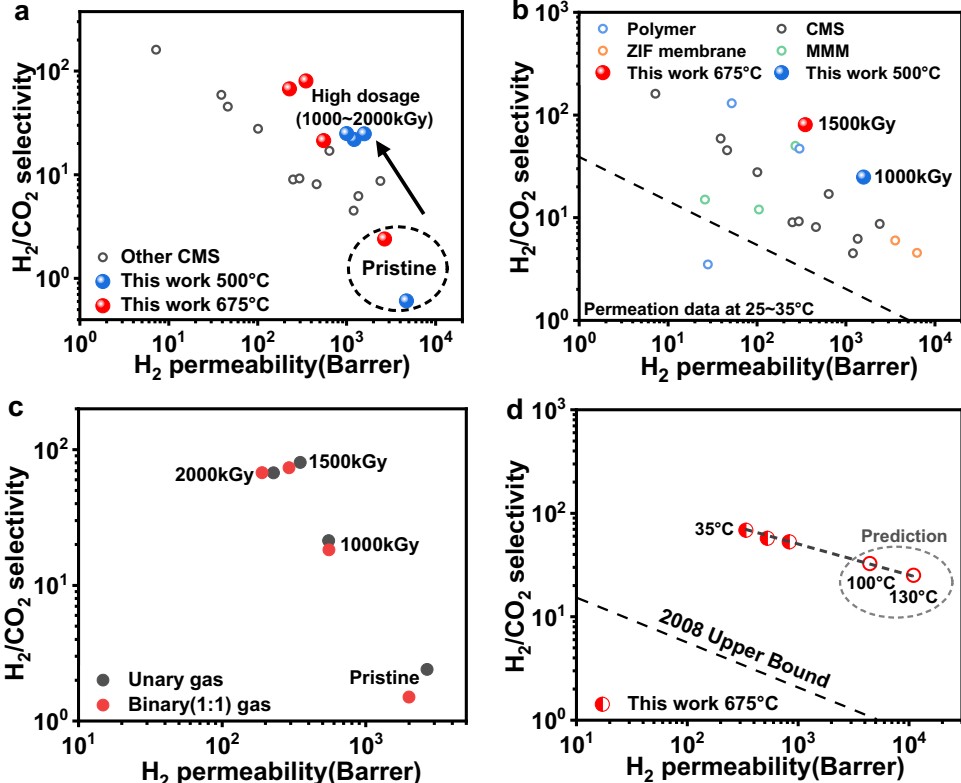

**Fig. 4 | Gas separation performances of the highly irradiated CMS membranes in H₂/CO₂ pair. a** H₂/CO₂ permeation data of pristine and highly electron-irradiated CMS membranes for two different temperatures (500 °C and 675 °C) including other state-of-the-art CMS membranes. **b** H₂/CO₂ permeation data of 500 °C 1000kGy- and 675 °C 1500kGy-CMS including other state-of-the-art membranes. **c** H₂/CO₂ separation performances of 675 °C CMS membranes in unary and binary (equimolar mixture) systems. **d** H₂/CO₂ experimental data of 675 °C 1500kGy-CMS membrane at permeation temperatures of 35 °C, 45 °C, and 55 °C and predictions at 100 °C and 130 °C. Source data are provided as a Source Data file.

negligible sorption selectivity influence on the improved separation capability, regardless of C₂H₄ and C₂H₆, almost the same isotherm curve and final uptake amount were observed in pristine and electron-irradiated CMS, respectively.

To explain the fast and selective ethylene transport in the electron-irradiated CMS membranes, the $sp^3$-/$sp^2$-hybridized carbon ratio for pristine and electron-irradiated CMS (50, 100, 250 kGy) samples were determined via XPS depth profiles (Fig. 3f). By deconvoluting the XPS C1s spectrum to three Gaussian peaks, the $sp^3$ and $sp^2$-hybridized carbon peaks were obtained, and the ratio was estimated by their peak area (Supplementary Figs. 19–22)[21]. It is generally known that the $sp^3$-hybridized carbon obstructs the packing of the strands due to the three-dimensional atomic configuration, which makes faster penetration of smaller gas molecules. In contrast, the two-dimensional $sp^2$-hybridized carbon is concerned with closely packed carbon strands, thus a more selective CMS structure for specific gas pairs. As shown in Fig. 3f, the $sp^3$/$sp^2$ ratio is consistent (~0.6) for all depths in pristine CMS, whereas electron-irradiated CMS shows the $sp^3$/$sp^2$ ratio gradient depending on the depth. At the surface, 50kGy-CMS, 100kGy-CMS and 250kGy-CMS had a low $sp^3$/$sp^2$ ratio (0.40, 0.35 and 0.28, respectively), and the ratio gradually increased to arrive at a similar value of the pristine CMS (~0.6) below 2.75 µm, which indicates that the penetrant molecules could undergo significant transport resistance on the electron-irradiated CMS surface. Therefore, the substantial portion of accessible ultra-micropores on the surface are both oxygen-functionalized and tightly packed, rejecting the permeation of C₂H₆ over C₂H₄.

We extended our finding in the unary permeation of electron-irradiated CMS to binary gas mixture permeation, revealing the possible effects of multicomponent interaction within the CMS membranes. For this, the Wicke-Kallenbach mode was used to test

equimolar C₂H₄/C₂H₆ mixtures (Supplementary Fig. 23). For the 50kGy-CMS, the permeability of C₂H₄ was 368.3 Barrer in the unary permeation, while that of mixed gas was 352.6 Barrer. Roughly a 4% loss in permeability was observed in this case, and the selectivity of the mixture permeation slightly decreased from 5.4 to 4.7 (Fig. 3g). The 100kGy-CMS and 250kGy-CMS showed comparable selectivity between unary (9.1 and 14.0 for 100kGy-CMS and 250kGy-CMS, respectively) and binary permeation (8.0 and 13.8 for 100kGy-CMS and 250kGy-CMS, respectively). For all CMS membranes tested, including pristine and electron-irradiated ones, a slight reduction (about 2–15%) in the mixture selectivity was noticed, which was mainly due to the coupling effect among different gas species within the ultra-micropores of the CMS membrane[41,42]. The equimolar binary permeation tests revealed more realistic performances of electron-irradiated CMS membranes under industrially relevant gas mixtures.

In the previous unary permeation tests, we noticed that when the electron irradiation dosage increased to the extreme level, exceeding 1000 kGy, almost complete blocking of the N₂ permeation (0.364 nm) while maintaining a moderate permeability of He (0.26 nm, Supplementary Fig. 12) was achieved. Motivated by this dramatic change in permeabilities at high dosages, we further broadened our sight to another key energy-intensive separation pair of H₂/CO₂ (Δsize = 0.04 nm) to investigate the versatility of our proposed membrane tuning method. Both unary and binary permeation tests were performed for CMS membranes prepared at 500 °C and 675 °C (Fig. 4a-c and Supplementary Fig. 24). As shown in Fig. 4a, the separation performances of both pristine CMS membranes are concentrated on the lower right corner of the Robeson plot, showing mediocre H₂ sieving capabilities of the CMS membranes—primarily due to the insufficient population of ultramicropore windows that are small enough to block

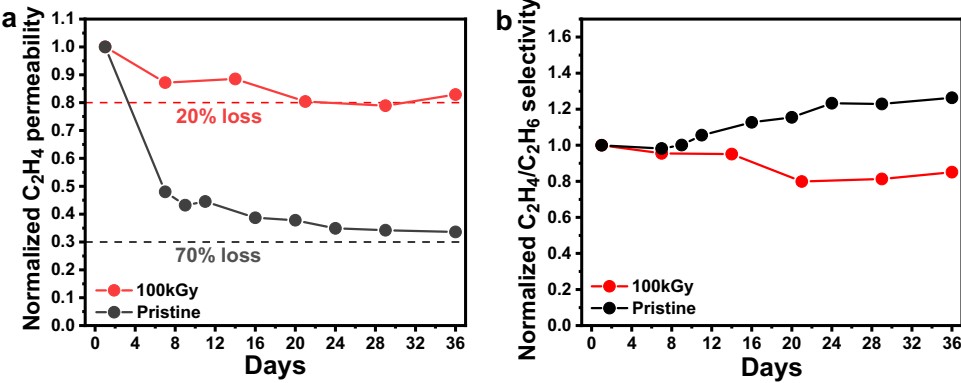

**Fig. 5 | Aging tracking of pristine- and 100kGy-CMS membranes. a, b** Normalized $C_2H_4$ permeability and $C_2H_4/C_2H_6$ selectivity data versus experimental time (days). Source data are provided as a Source Data file.

the penetration of $CO_2$ as well as the strong sorption of $CO_2$ onto the micropores of the CMS. With high-dosage irradiation (1000–2000 kGy) of those CMS membranes, notable performance enhancements were observed toward making highly $H_2$ selective CMS membranes. In the case of the CMS membrane pyrolyzed at 500 °C, with 1000 kGy irradiation, ideal $H_2/CO_2$ selectivity was increased by a factor of 40. Remarkably $H_2$-selective membranes were formed when CMS membranes pyrolyzed at 675 °C were irradiated with 1500 kGy, showing outstanding unary $H_2$ permeability over 350 Barrer and ideal $H_2/CO_2$ selectivity over 80. Figure 4b compares the separation performances of 1000kGy-CMS (pyrolysis at 500 °C) and 1500kGy-CMS (pyrolysis at 675 °C) membranes in the Robeson plot, revealing the superior molecular sieving properties of the high-dosage electron-irradiated CMS membranes (Supplementary Tables 5–8).

Binary gas permeation tests were also completed for a $H_2/CO_2$ pair to elucidate the multicomponent permeation effects (Fig. 4c). The binary permeation of the pristine CMS membrane (pyrolysis at 675 °C) showed quite a large deviation from unary permeation, especially with a noticeable decrease in both $H_2$ permeability and $H_2/CO_2$ selectivity. Favorable and competitive $CO_2$ adsorption over $H_2$ of the micropores of the CMS membrane is a plausible cause of this result[43]. On the other hand, both 1000kGy-CMS and 1500kGy-CMS showed largely unchanged separation performances for binary permeation, which might have been caused by the decrease of the population of ultramicropores that could favorably accommodate $CO_2$ molecules in those heavily irradiated membranes, which reduced the impact of competitive adsorption for binary permeation. Notably, binary permeation in 2000kGy-CMS showed a marginal permeability loss among other irradiated samples, with rather increased selectivity than ideal selectivity. This powerful molecular sieving would have resulted from the further removal of transport pathways favorable to larger $CO_2$ molecules in massively irradiated CMS membranes. We further confirmed the superior $H_2/CO_2$ separation performances of electron-irradiated CMS membranes at high temperatures covering from 100 °C to 130 °C, the realistic feed condition directly coupled with industrial water-gas shift reactors. Experimental data (1500kGy-CMS) from permeation tests at 35 °C, 45 °C, and 55 °C were used to calculate the apparent activation energy of permeation using Arrhenius regression (Fig. 4d). We expected that the permeability of the electron-irradiated CMS membranes would significantly increase at such high temperatures, keeping slightly reduced $H_2/CO_2$ selectivities, which is still a remarkable separation performance.

## Aging resistance of electron-irradiated CMS

Physical aging generally occurs in non-equilibrium state glassy polymers. It entails a segmental rearrangement process that leads the polymer with excess free volume to densification and then further approaches the thermodynamically stable state over time[30,44]. This self-retarding process was also reported in CMS membranes and significantly degraded the separation performance for long-term operation[45,46]. To investigate the effect of aging on electron-irradiated CMS membranes, both pristine CMS and 100kGy-CMS were tested for a $C_2H_4/C_2H_6$ pair. We monitored the change of $C_2H_4$ permeability and $C_2H_4/C_2H_6$ selectivity for up to 36 days (Fig. 5). The pristine CMS membranes immediately lost 60% of the initial permeability within 6 days. They converged to only 30% remaining permeability after 36 days, whereas the selectivity gradually increased to 120%, which is a typical aging behavior for CMS membranes. In contrast, 100kGy-CMS preserved more than 80% of $C_2H_4$ permeability during the same period (36 days), and unlike the pristine sample, a slight decrease (~20%) in selectivity was observed, showing excellent aging resistance. 87 K Ar physisorption experiments (Supplementary Figs. 25 and 26) demonstrated the relatively reduced pore volume (pristine: 0.286 $cm^3/g$; 100kGy-CMS: 0.269 $cm^3/g$) and BET surface area (pristine: 733.9 $m^2/g$ and 100kGy-CMS: 691.9 $m^2/g$) in 100kGy-CMS, implying the relaxation of the excess free volumes in CMS. Moreover, the previously discussed structure characterization, such as TEM and GI-WAXS (Figs. 1 and 2), also indicated that the electron irradiation tuning reduced the ultramicropore dimensions. The reductions in pore volume can illustrate relatively small $C_2H_4$ permeability loss in 100kGy-CMS since the already tightened CMS structure less induce pore shrinkage. Additionally, elemental analysis (EA) was performed to figure out the aging characteristic of the CMS membranes (Supplementary Table 9). The aged pristine CMS (6.92%) revealed a relatively higher oxygen fraction than that of fresh pristine CMS (6.68%) because reactive carbon edges slowly chemisorb oxygen in air condition[47,48]. The oxygen fraction in 100kGy-CMS decrease from 9.95% to 9.06% in 30 days. This result is comparable to the loss of the oxygen-containing group in aged graphene oxide (GO) and can be considered as a process where over-oxidized CMS surface rebound back to equilibrium structure[49]. The unusual aging behavior (selectivity decrease) of the electron-irradiated CMS is conceivable with hypothetical scheme based on the changes in oxygen fraction of the 100kGy-CMS after aging process (Supplementary Fig. 27). The $D_1$ and $D'_1$ are supermicropore dimensions and $D_2$, $D_3$, $D'_2$ and $D'_3$ are ultramicropore of the fresh and aged electron-irradiated CMS, respectively, according to the IUPAC definition. When aging process proceeds, the supermicropore dimension $D_1$ shrinks to $D'_1$ and entails overall permeability reduction in both $C_2H_4$ and $C_2H_6$. Oxygen adsorbed on the carbon edge sites is less developed after aging and it makes ultramicropore dimensions ($D'_2$ and $D'_3$) increased than the fresh sample, resulting in slightly lessened selectivity of the aged CMS. That is, contrast structural evolution in supermicropore and ultramicropore explains the unique permeation behavior of 100kGy-CMS after aging process.

## Discussion

In summary, we successfully demonstrated that post-pyrolysis electron irradiation tuned the microstructure of 6FDA:BPDA(1:1)-DAM derived CMS membranes by precisely adjusting the irradiation dosage. During electron irradiation, the generated carbon defects induced ultramicropore tightening and oxidized surface formation on the CMS. The electron-tailored ultramicropore size distribution was shifted to be more concentrated in the sub-0.4 nm range, which fits well for discriminating small hydrocarbon gases. As the electron dosage increased, the size-sieving effect of the membrane enormously increased, which in turn enhanced aging resistance. The binary gas permeation test on the studied gas pairs ($C_2H_4/C_2H_6$ and $H_2/CO_2$) showed comparable separation capabilities to that of unary gas test and more realistic performances of electron-irradiated CMS membranes. $H_2/CO_2$ performances at high temperature (100 and 130 °C) were also calculated to confirm separation behavior and viability of the electron-irradiated CMS, indicating remarkable performance preservation in industrially relevant conditions. In the perspective of post-treatment of nanoporous carbon membranes, electron-irradiation shares similarities with the previously reported hyperaging process[19]. Both methods intentionally reduce the distance between carbon strands or plates, decreasing the free volume of the membrane to the equilibrium state. However, electron irradiation is not a time-dependent thermal treatment and, more importantly, it combines the advantages of both surface oxidation and pore tightening. Compared to other surface treatment methods, ozone treatment to Matrimid® derived CMS was reported by purging ozone to CMS at room temperature[50]. As with the case of $O_2$ doping treatment, ozone-modified CMS shows similar performance behavior, namely, a selectivity increase and a permeability decrease. However, an insignificant tuning degree of the modified CMS does not create an attractive separation performance for challenging gas pairs such as $H_2/CO_2$. In this regard, it can be acknowledged that electron irradiation opens a stage for an attractive tuning factor distinguished from the conventional CMS tuning, which is mainly limited to thermal treatment.

## Methods

### Materials

4,4-(Hexafluoroisopropylidene)diphthalic Anhydride(6FDA, 98.0%, TCI chemicals), 2,4,6-trimethyl-1,3-phenylenediamine (DAM, 98.5%, TCI chemicals) and 4,4′-Biphthalic Anhydride (BPDA, 98%, TCI chemicals) were sublimated in vacuum state before use. Acetic anhydride (AcAn, >99%, Sigma-Aldrich) and 3-methylpyridine (beta picoline, 99%, Sigma-Aldrich) were dried by soaking 3 A zeolite sieve before use. N-methyl-2-pyrrolidone (NMP, 99%, Alfa Aesar), methanol(MeOH, 99.5%, SAMCHUN) and n-hexane(95.0%, SAMCHUN) was used as received. The used gases (>99.9%) in this experiment were provided by Special Gas, Ltd (Korea) including helium(He), carbon dioxide ($CO_2$), nitrogen($N_2$), methane($CH_4$), ethylene($C_2H_4$) and ethane($C_2H_6$).

### 6FDA:BPDA(1:1)-DAM synthesis

6FDA:BPDA(1:1)-DAM was synthesized by two step polycondensation reaction, the conventional polyimide synthesis method. All the monomers were sublimated in vacuum state and the solvents were dried by 3 A zeolite sieve to remove residual moisture. For the first step, the stoichiometric weighed diamine(DAM) and dianhydrides(6FDA and BPDA) monomers were dissolved in NMP under low temperature (<5 °C) and $N_2$ purging condition to produce polyamic acid for 1 day. Then chemical imidization was proceeded to convert polyamic acid to cyclic polyimide at ambient temperature, using beta picoline as a catalyst and AcAn as a dehydrating agent respectively. The produced polyimide was soaked in MeOH and dried at 210 °C in vacuum oven.

### Polymer precursor film fabrication

To remove the moisture, the polymer powder was prepared in vacuum oven at 120 °C for 12 h before use. 23 wt% polymer solution was formed by dissolving the dried powder in NMP and placed on a roller for homogeneous quality. The polymer solution, film applicator and a glass plate and beak containing NMP were prepared in glove bag. Then the bag was purged with $N_2$ two or three times, sealed and saturated with NMP for slow solvent evaporation in solution. Next, the solution was cast on a glass plate with film applicator and evaporated thoroughly for 1 day. The polymer film was removed from the plate and placed in DI water, MeOH and n-hexane sequentially for solvent exchange. Afterwards, the polymer film was dried in 120 °C vacuum oven for overnight.

### Spin coated film fabrication for GIWAXS analysis

5–8 wt% polymer-NMP solution was prepared and spin-coated on the surface of silicon wafer at 6000 rpm for 60 s. After the coating process, the coated wafer samples were moved immediately into the NMP saturated glove bag for thin and dense film formation. The resultant films had thickness approximately below 1 μm.

### Grazing incidence wide angle X-ray scattering (GIWAXS) analysis

Grazing incidence wide angle X-ray scattering (GIWAXS) measurement was performed on beamline 9 A(U-SAXS) in Pohang Accelerator Laboratory (Pohang, Korea) with $E_k$ = 11.07 keV and a wavelength of 1.101 Å. The beam incidence angle was set in the range 0.12–0.15° which is between the critical angle of the sample (polymer or CMS) film and the silicon substrate. The scattered beams from thin film were recorded by a two-dimensional CCD detector, a sample to detector distance of 0.22 m.

### CMS membrane fabrication (pyrolysis protocol)

The polymer films were placed on a corrugated quartz plate and the plate was loaded in three-zone tube furnace (SH scientific, SH-FU-80LTG-0M). Providing ultra-high purity (UHP, 99.999%) argon through mass flow meter (Omega Engineering), oxygen level inside the quartz tube was maintained around 30ppm prior to pyrolysis. Then the polymer films were soaked at 500 °C, 600 °C and 675 °C for 2 h (Ramping protocol; RT~250 °C: 10 °C/min, 250~X-20 °C: 3 °C/min, X-20~X °C: 0.2 °C/min, 2 hr soaking at X °C, X=500 °C, 600 °C and 675 °C).

### Electron irradiation

Electron beam was irradiated on the fabricated CMS membranes using conveyor type electron accelerator No.4 at EB-Tech Co., Ltd. (Daejeon, Korea) The samples were placed on thin aluminum foils and the foils were taped to steel plate stage to hold the sample. Each turn was irradiated by 25kGy, and the number of turns was adjusted according to the final irradiation dosage (For 1 turn, beam energy: 1 MeV, current: 9.3 mA, conveyor speed: 10 m/min).

### Single gas permeation

First, the prepared CMS membrane was loaded on permeation cell. Permeation test was carried out with pure single gas, He, $H_2$, $CO_2$, $N_2$, $CH_4$, $C_2H_4$ and $C_2H_6$ at 35 °C under 2 bar upstream pressure using constant volume method. The downstream was kept under vacuum state before measurement start. When the feed is provided from the upstream and passes the membrane, the pressure transducer detects the pressure rise in downstream volume. The pressure rise rate was utilized to calculate the permeability of each gas (Supplementary Fig. 8).

### Mixed gas permeation test

Schematic diagram of mixed-gas permeation test is shown in Supplementary Fig. 23. The prepared CMS membranes was mounted on the membrane cell and temperature was maintained at 35 °C. By mass flow

controller (5850E, BROOKS), the equimolar $C_2H_4/C_2H_6$ mixture feed and Helium sweep gas were provided to the membrane cell with a flow rate of 15sccm respectively. The rejected feed was vented to retentate side and the permeate penetrants were swept by He to gas chromatography (YL6500 GC, Young Lin Instruments) equipped with flame ionization detector and thermal conductivity detector. In the case of $H_2/CO_2$ separation, the equimolar $H_2/CO_2$ mixed gas was used as feed gas and Ar was flew as sweep gas to push the permeated gases to GC. For GC quantitative evaluation, additional calibrations using standard samples with various known concentration were carried out.

**Pressure decay sorption test**

For custom-built pressure decaying system, the fabricated CMS film was smashed into small pieces and loaded into a sample tube. Then the tube was wrapped with heating wire to evacuate the system inside and degas the sample prior to test. After the evacuating process, the system was placed in a water bath with a thermostat to preserve uniform temperature (35 °C). The gas reservoir was filled with target gas, equilibrating the temperature for about 10 min, then the target gas was introduced into the sample tube by controlling the valves. The pressure of the reservoir and the sample tube was measured via a high-resolution pressure transducer. (Supplementary Fig. 14).

## Data availability

All data supporting the results of this study are available within the article and its supplementary information file. Source data are provided with this paper.

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

## Acknowledgements

This research was supported by Basic Science research program through the National Research Foundation of Korea (NRF) funded by Ministry of Science, ICT & Future Planning (No. NRF-2021R1C1C1012014). This work was also supported by Samsung Research Funding & Incubation Center of Samsung Electronics (SRFC-MA1902-08). The GI-WAXS analysis in the work was performed at 9 A beam lines of the Pohang Accelerator Laboratory, Korea.

## Author contributions

B.O., H.S. and D.-Y.K. conceived the concept of the research and implemented the experiment. B.O. carried out the unary and binary gas permeation test, pressure decay experiment, TEM, XPS, 87 K Ar physisorption and elemental analysis. B.O. and H.S. performed 6FDA:BPDA(1:1)-DAM polyimide synthesis and Raman analysis. B.O. and J.C. performed SEM, EDS and GIWAXS analysis. B.O. and D.-Y.K wrote the manuscript. H.S., J.C. and S.L. helped to revise the manuscript. All authors contributed development of research discussion and approved the final manuscript.

## Competing interests

The authors declare no competing interests.
