## [Peer Review File · Nature Communications]

Reviewers' comments:

Reviewer #1 (Remarks to the Author):

The authors report 10 μm thin Carbon Molecular Sieve (CMS) membranes synthesized from polymeric membranes as supporting and assisting layer, which is here chosen to be 6FDA:BPDA(1:1)-18 DAM. As a novel method to previously reported studies they chose electron radiation for controlled pyrolysis of the polymer to graphene-like carbon flakes and turbostratic carbon, that are staggered in a way to sieve molecules in the ultramicroporous range. The main point of this paper is the C₂ separation, which is challenging due to the 0.015 nm kinetic size difference between ethylene/ethane. The authors must have spent years to collect all these data, but there are several concerns to raise upon the generated data and some general misconceptions.

Concerns regarding the material and its characterization:

1) Page 6, Column 108-115: I think the explanation through chain-packing and fractional free volume needs some more explanation and evaluation. The separation mechanism and flux through a polymeric membrane is described by the solution-diffusion model and cannot be described by molecular sieving, even though this is the case for the graphene filler. There are also other effects in a polymeric membrane accounting into flux and separation values, such as swelling, aging, fouling.

2) The comparison of 6-FDA-DAM co-polymers with Matrimid is not very smart since both polymers have already different application fields. While 6-FDA-DAM co-polymers are widely applied for Hydrocarbon Separation, Matrimid is not.

3) Figure 1b,c,d: The images should be replaced as SEM is not the method needed here for showing the CMS layer, the magnification and contrast is not high enough. I would suggest ultramicrotome cuttings of your membrane for TEM measurements in scanning mode. To see the CMS film on-top of the polymer, STEM allows a much better look in the cross-section of the film. I am not able to follow your conclusions drawn from here.

4) Figure 1e,f,g,h: The TEM images look exactly the same, but they are taken from different spots/samples. However, I do not see any difference after all, both look like turbostratic carbon. Could the authors please comment on the electron diffraction more? Is it even possible to distinguish those two layers by electron diffraction with the resolution of your TEM? For me the question is: Does it make sense to show these images, whilst the difference is not visible from them?

5) SI Figure 2: Oxygen in EDS is nothing you could easily quantify. The error in light elements with the EDS detector is probably higher than the difference measured here. The noise of the line scan is extremely high. Please comment on this measurement and the quality of the SEM data.

6) Figure 2: The oxygen doping of the CMS layer is clearly visible in the depth-profile, however, the data e.g. Raman, XPS is hardly visible. The data needs to be placed in its raw state into the SI. Here, y-axis shifting and arbitrary units make it unable to draw conclusions. I do not see the differences here as they are very small. I am unable to follow your conclusions because I cannot see differences in the plots. Please also comment in this regard on SI Figure 13, 14, 15 where you show tiny differences in the C-hybridization – how reliable is this quantification.

Regarding Permeation Data:

i) What is the contribution of the underlying polymeric membrane to your separation values?

ii) I understand the method of making a tiny sample for pressure-decay method in very small scale. How were you able to electron-beam radiate a whole sample for Wicke-Kallenbach measurements? How long did it take?

iii) To compare your data to the benchmarks with 6-FDA-DAM copolymers, I am missing the pressure, at which you measured your membrane. At one point you say 2 atm. Literature most of

the time uses 35°C and 3-4 atm, there is no statement on why you change the pressure, and if its 2 atm only, the comparison to the benchmark cannot be drawn.

iv) In order to proof molecular sieving for the samples, single gas permeances of He, H₂, N₂, CO₂, CH₄, C₂, C₃, must be performed. In a permeability vs. kinetic gas diameter plot of the single gas permeances, a sharp cut-off will be visible.

v) Figure 5: The loss of the ethylene permeability is due to aging effects of the polymer. The polymer is still the biggest part of your membrane after 100kGy treatment. How is it possible that you lose much less permeability with the turbostratic graphite on top, while the selectivity goes up? It would be favourable to show the permeability loss of the ethane as well, because both data sets are used to calculate the selectivity. Also: Show the raw data and not a normalization. If it can be seen whether the CMS membrane has much lower permeability, it should become obvious why the aging is slower and why it should not be interpreted in comparison.

vi) The whole part of the aging resistance due to the CMS layer is based on wrong assumptions. The polymer is still 99% of the membrane and therefore, the aging must be the same.

vii) How is the reproducibility guaranteed? Please provide tables of the measured gas permeabilities for several membrane samples and include error bars to your data points.

Reviewer #2 (Remarks to the Author):

General Comments.

Oh et al. reported on the electron-irradiated CMS membranes for enhanced H₂/CO₂ and C₂H₄/C₂H₆ separation performance. Although it is well-known that the structure of carbon materials can be engineered by high-energy electron irradiation, the authors nicely demonstrated the structure-separation performance relationship of the CMS membranes. However, there are still some issues that the authors should address before publication. The detailed comments are described below.

Comment 1:

In lines 48-50 on pg 2, the authors mentioned that the limited separation performance was attributed to insufficient pore window curvature, which is not clear.

Comment 2:

The authors should justify the selection of 6FDA-BPDA:DAM (1:1) for the polymer precursor. Also, specify the thickness of the films used in this work.

Comment 3:

The images in Figure 3a is not very clear to understand how the electron irradiation changed the ultramicropores of CMS membranes. The authors should explain the mechanism of changing the ultramicropores of CMS membranes via electron irradiation.

Comment 4:

The sp³/sp² ratio of the 250 kGy-CMS membrane is still lower than that of the 100 kGy-CMS analog even at 3.75 μm in depth (Figure 3f), reflecting that the 250 kGy dosage is still strong enough to affect the structure of CMS membranes at 3.75 μm in depth. Also, the higher electron beam intensity is expected to increase the effective thickness available for the electron irradiation as well as the sp²/sp³ ratio? If so, the authors need to provide the effective thickness for the electron irradiation as a function of the number of irradiation or the electron beam intensity.

Comment 5:

The electron-irradiated CMS membrane is sort of asymmetric membrane since the effective thickness available for the electron irradiation is limited; however, the current analysis on the transport result is based on symmetric dense film structure, which is wrong. The authors should analyze the transport results by using a series model by assuming that the remaining electron-irradiation irrelevant CMS membrane would show the same separation performance as that for the pristine CMS analog, which can allow them to induce the intrinsic separation performance of the electron-irradiated CMS membranes. This comment is partly related to the previous comment in #4 (the effective thickness for the electron irradiation as a function of the electron beam intensity).

Comment 6:

The authors should explain why the C₂H₄/C₂H₆ selectivity of the 100 kGy-CMS membrane decreased after 36 d aging process (Figure 5b). In fact, the explanation on the enhanced aging resistance of the electron-irradiated CMS membrane is missing.

Comment 7:

The mechanical integrity is an important issue in developing carbon membranes for gas separation. The authors should provide the mechanical property of the electron-irradiated CMS membranes and compare it with that of the precursor as well as the pristine CMS membrane.

Reviewer #3 (Remarks to the Author):

The article entitled "Precision Carbon Membrane: Electron-mediated Control of Nanoporosity for Targeted Molecular Separation" presents the significance of carbon molecular sieves supported polymeric membrane for the gas separation applications. on preparation of charged Ultrafiltration membranes. It gives the detailed information about the physicochemical characterization and selectivity analysis of C₂H₄/C₂H₆ and H₂/CO₂. The article is well written, however minor revision is needed before this paper can be considered for the publication.

1. In abstract, the pronouns could be avoided and the selectivity data can be provided.
2. In morphology and XPS analysis (Fig.2e), there is no salient difference in pristine bean and electron beam irradiated membrane. It could be explained in detail.
3. The mechanism of CMS deposition and membrane formation need to discuss with literature. The thickness of deposited layer could be included and the influence of gas separation could be drafted.
4. The stability of membrane before and after modification need to be discuss in detail with references. Plasticization effect of membrane with few characterizations could also be discussed.
5. Schematic of CMS on selective gas permeation can be included.
6. Scope of studies and viability of gas separation could have discussed.

Revision Notes to: “Precision Carbon Membrane: Electron-mediated Control of Nanoporosity for Targeted Molecular Separation”

Manuscript ID: NCOMMS-22-08914

Reviewer #1

The authors report 10 μm thin Carbon Molecular Sieve (CMS) membranes synthesized from polymeric membranes as supporting and assisting layer, which is here chosen to be 6FDA:BPDA(1:1)- DAM. As a novel method to previously reported studies they chose electron radiation for controlled pyrolysis of the polymer to graphene-like carbon flakes and turbostratic carbon, that are staggered in a way to sieve molecules in the ultramicroporous range. The main point of this paper is the C₂ separation, which is challenging due to the 0.015 nm kinetic size difference between ethylene/ethane. The authors must have spent years to collect all these data, but there are several concerns to raise upon the generated data and some general misconceptions.

The reviewer gave excellent suggestions in a number of areas that greatly strengthened the conclusion of the article. However, we have discovered that there are certain misunderstandings that may have caused by our explanations. The carbon molecular sieve (CMS) membrane employed in our study was compared to a thin-film composite (TFC) membrane by Reviewer #1, who understood that the separate CMS layer was formed on the polymer support. Our CMS membrane, on the other hand, is a symmetric, free-standing membrane fabricated without the use of any support elements. Furthermore, electron irradiation was applied as a post-pyrolysis treatment as a separate experiment. For a better understanding, a schematic illustration for the CMS fabrication and electron irradiation process from polymer film is included here.

Figure for Review. Schematic for CMS fabrication from symmetric polymer film and electron irradiation process on symmetric CMS film.

Concerns regarding the material and its characterization:

1) Page 6, Column 108-115: I think the explanation through chain-packing and fractional free volume needs some more explanation and evaluation. The separation mechanism and flux through a polymeric membrane is described by the solution-diffusion model and cannot be described by molecular sieving, even though this is the case for the graphene filler. There are also other effects in a polymeric membrane accounting into flux and separation values, such as swelling, aging, fouling.

As the reviewer pointed out, the solution-diffusion model would explain the separation mechanism of the both polymeric and CMS membrane. The diffusion of penetrant molecules in the polymer matrix is connected to chain-packing and fractional free volume (FFV), but as stated by the reviewer, this may not be sufficient to fully explain the separation performance of 6FDA-based polyimide. Therefore, the parenthetical below has been added to the main body.

- Page 6, Lines 106-108: 6FDA-based polyimides have shown high gas separation performances and resistance to swelling due to structural rigidity and inefficient chain packing from trifluoromethyl group (-CF₃) on the backbone.^{28,29}

The following new references were added

28. Roy, P. K. *et al.* Investigations on 6FDA/BPDA-DAM polymer melt properties and CO₂ adsorption using molecular dynamics simulations. *J. Memb. Sci.* **613**, 118377 (2020).

29. Rungta, M. *et al.* Carbon molecular sieve structure development and membrane performance relationships. *Carbon N. Y.* **115**, 237–248 (2017).

2) The comparison of 6-FDA-DAM co-polymers with Matrimid is not very smart since both polymers have already different application fields. While 6-FDA-DAM co-polymers are widely applied for Hydrocarbon Separation, Matrimid is not.

According to the reviewer, it is not a good comparative example because the application fields of Matrimid and 6FDA-based polyimide are different. As a result, the direct comparison between 6FDA-based polyimide with Matrimid is removed and the following parenthetical has been inserted to the main body

- Page 6, Lines 111-113: Therefore, carbon molecular sieve membranes fabricated with 6FDA-based polyimides indicated superior permeability for gas separation, especially in light hydrocarbons, than other commercialized or studied precursor derived CMS.³⁰⁻³³

The following new references were added

32. Hazazi, K. *et al.* Ultra-selective carbon molecular sieve membranes for natural gas separations based on a carbon-rich intrinsically microporous polyimide precursor. *J. Memb. Sci.* **585**, 1–9 (2019).

33. Chu, Y.-H. *et al.* Iron-containing carbon molecular sieve membranes for advanced olefin/paraffin separations. *J. Memb. Sci.* **548**, 609–620 (2018).

3) Figure 1b,c,d: The images should be replaced as SEM is not the method needed here for showing the CMS layer, the magnification and contrast is not high enough. I would suggest

ultramicrotome cuttings of your membrane for TEM measurements in scanning mode. To see the CMS film on-top of the polymer, STEM allows a much better look in the cross-section of the film. I am not able to follow your conclusions drawn from here.

Both the pristine- and 250kGy-CMS films in **Figure 1b-d** were symmetric and free-standing, with no support (please refer to the below SEM images showing symmetry of our membrane). As a result, it is not unexpected that the SEM images did not show “CMS film on-top of the polymer” as a separate layer. SEM analyses was originally intended for the demonstration of the consistent surface morphology of symmetric CMS membranes before and after electron irradiation. The following parenthetical has been modified to the main body.

- **Page 7, Lines 118-120: The CMS were prepared as self-standing and symmetric films then irradiated with a range of 50 – 250 kGy by a high-energy electron-beam accelerator in ambient air.**

Figure for Review. SEM images with high magnification of pristine-CMS(up) and 250kGy-CMS(down)

4) Figure 1e,f,g,h: The TEM images look exactly the same, but they are taken from different spots/samples. However, I do not see any difference after all, both look like turbostratic carbon. Could the authors please comment on the electron diffraction more? Is it even possible to distinguish those two layers by electron diffraction with the resolution of your TEM? For me the question is: Does it make sense to show these images, whilst the difference is not visible from them?

This comment appears to be based on the same misconception as the previous one. **Figure 1e-g** show the turbostratic structure of the pristine- and 250kGy-CMS, as well as their FFT pattern (**Figure 1f** and **1h**). **Figure 1e** and **1g** show that the pristine and 250kGy-CMS films have essentially identical maze-like structures, indicating that electron irradiation has little effect on the macro morphologies such as amorphous and disordered features of CMS films. However, after electron irradiation, FFT patterns show the development of compact and denser carbon structure in micro-scale.

5) SI Figure 2: Oxygen in EDS is nothing you could easily quantify. The error in light elements with the EDS detector is probably higher than the difference measured here. The noise of the line scan is extremely high. Please comment on this measurement and the quality of the SEM data.

We understand your concern about using EDS to quantify light elements. Instead of quantifying the oxygen composition of the electron irradiated CMS, the EDS line scan was employed in the revised manuscript to show the tendency of oxygen enrichment on the CMS surface. The oxidized layer with a thickness of roughly $4\mu\text{m}$ was also proven by XPS depth profiling of the 500kGy-CMS (**Figure S5**), which was confirmed by the EDS line scan. Because EDS line scan noise is severe, the resolution was increased to $0.1\ \mu\text{m}$ and the number of frames was maximized as well.

Figure S2 has been updated:

Supplementary Fig. 2. Oxygen line scan from Energy-dispersive X-ray spectrometer (EDS), indicating oxygen versus CMS depth (10mm) of 500°C pristine (left)- and 500kGy (right)-CMS.

6) Figure 2: The oxygen doping of the CMS layer is clearly visible in the depth-profile, however, the data e.g. Raman, XPS is hardly visible. The data needs to be placed in its raw state into the SI. Here, y-axis shifting and arbitrary units make it unable to draw conclusions. I do not see the differences here as they are very small. I am unable to follow your conclusions because I cannot see differences in the plots. Please also comment in this regard on SI Figure 13, 14, 15 where you show tiny differences in the C-hybridization – how reliable is this quantification.

XPS survey scans and Raman spectra without arbitrary unit were included to Supporting Information, as suggested by the reviewer (**Figure S4** and **S7**). XPS survey scans of pristine- and 250kGy-CMS was plotted separately (black for pristine and red for 250kGy-CMS) and y-axis indicates counts/s for corresponding binding energy. When compared to pristine CMS, the O1s peak near 530 eV is noticeable for 250kGy-CMS (**Figure S4**). Furthermore, Raman spectra for pristine and 250kGy-CMS were overlaid without arbitrary units, and the intensity values for D and G peak, as well as I_D/I_G ratios were also shown in a table. (**Figure S7**).

Figure S4 has been added

Supplementary Fig. 4. X-ray photoelectron spectroscopy (XPS) survey scan of the 500 °C pristine (left)- and 250kGy (right)-CMS without y-axis arbitrary unit (Counts/s vs Binding energy). The intensified O1s peak around 530eV become clear after electron irradiation.

Figure S7 has been added.

	Pristine-CMS	250kGy-CMS
I_D	4791.224	4214.635
I_G	7182.265	6778.74
I_D/I_G ratio	0.667	0.621

Supplementary Fig. 7. Raman spectrum of the 500 °C pristine- and 250kGy-CMS as a version without arbitrary unit (above). The below table shows the maximum value of D and G peak at each CMS, which gives the calculated I_D/I_G ratio.

Figure S13-15 (new **Figure S19-22** in revised manuscript) shows XPS C1s peak deconvolutions for pristine-, 100kGy- and 250kGy-CMS (the deconvolution results of the different etched depths -- 0, 0.917, 1.834, 2.751, and 3.668 μm). The peak deconvolution may vary depending on the initial fitting parameters, we agree with the reviewer's concerns about reliable quantification in some circumstance. To minimize this uncertainty, peak fitting used the predefined x values (i.e., binding energy) for the C-O, sp^3 C, and sp^2 C peaks. C1s peak shapes (black line) at different depths in **Figure S19** is nearly identical due to homogeneity of the pristine CMS film. Therefore, the information from the deconvoluted peaks (C-O, sp^2 and sp^3) and constant area ratio of sp^3/sp^2 in pristine CMS are logically consistent. **Figure S20-22** shows that the C-O peak (287.8 eV) develops substantially at the surface of the CMS with a comparatively reduced sp^3 C peak (~ 285 eV). As the irradiation dosage increases, the C-O peak becomes more prominent (from 50 to 250 kGy).

Regarding Permeation Data:

i) What is the contribution of the underlying polymeric membrane to your separation values?

Since our CMS membranes are symmetric films, we will consider the “underlying polymeric membrane” in comment i) as “precursor polymeric membrane” before pyrolysis. The CMS membrane formed from 6FDA:BPDA(1:1)-DAM provides open but selective transport pathways for the penetrating molecules, similar to the non-pyrolyzed precursor membrane. We additionally performed gas permeation test on the precursor membrane for $\text{C}_2\text{H}_4/\text{C}_2\text{H}_6$ and H_2/CO_2 gas pairs and added the separation capabilities of the precursor to the separation performance plot.

Figure S11 has been added.

Supplementary Fig. 11. C_2H_4 permeability vs $\text{C}_2\text{H}_4/\text{C}_2\text{H}_6$ selectivity data for 500 °C pristine- and electron-irradiated CMS membranes compared with polymeric precursor membrane performance.

Figure S24 has been added.

Supplementary Fig. 24. H₂/CO₂ permeation data of pristine and highly electron-irradiated CMS membranes for two different temperatures (500 °C and 675 °C) including polymeric precursor membrane

ii) I understand the method of making a tiny sample for pressure-decay method in very small scale. How were you able to electron-beam radiate a whole sample for Wicke-Kallenbach measurements? How long did it take?

Scalability is one of the benefits of electron irradiation technique since it is already used in commercial large-scale applications including sterilization and semiconductor fabrication. In our work, the pristine CMS films (> 200 cm²) were placed on a sample stage with a size of 80cm*80cm and then irradiated at a controlled dosage inside the electron beam accelerator. As noted in the Supporting Information, the accelerator was equipped with conveyor belt containing sample stage and the dosage for one cycle was set at 25 kGy. The dosing of 500 kGy takes around 1 hour. For the gas permeation test (Wicke-Kallenbach measurement), we prepared the electron-irradiated CMS films with total area of 96 cm². For other analysis including physisorption and spectroscopic analyses, irradiated CMS films were broken down into small CMS fragments.

iii) To compare your data to the benchmarks with 6-FDA-DAM copolymers, I am missing the pressure, at which you measured your membrane. At one point you say 2 atm. Literature most of the time uses 35°C and 3-4 atm, there is no statement on why you change the pressure, and if its 2 atm only, the comparison to the benchmark cannot be drawn.

For single component permeation, we used 2 bar upstream pressure and for mixed gas permeation, 1 bar upstream pressure was used. According to the Supporting Information **Table S1-8**, roughly half of the previous literature conducted experiments at 3-4 atm, whereas other experimental conditions span from 1 to 7 atm.

iv) In order to proof molecular sieving for the samples, single gas permeances of He, H₂, N₂, CO₂, CH₄, C₂, C₃, must be performed. In a permeability vs. kinetic gas diameter plot of the single gas permeances, a sharp cut-off will be visible.

We additionally measured permeabilities for H₂, C₃H₆ and C₃H₈ of the pristine, 50kGy-, and 100kGy- to 250kGy-CMS to demonstrate the sharp cut-off of the electron irradiated CMS. H₂ with a kinetic diameter less than 3Å showed a modest permeability drop (4707.1 Barrer to 2847.4 Barrer as the electron dosage increases up to 250 kGy). In contrast to C₂H₄/C₂H₆ gas pair, the C₃H₈ permeability dropped dramatically for 50kGy-CMS, resulting in highly selective C₃H₆/C₃H₈ separation (selectivity increased from 13.28 to 47.13). When the irradiation dose is increased to 100kGy, the permeability for C₃H₆ dropped exponentially (by 32 times), indicating the almost complete blockage of the pores permeable to C₃ compounds. Due to the detection limit (0.72 Barrer) of the current setup, C₃ permeation data in 250kGy-CMS membrane could not be measured.

Figure 3b has been updated.

The following parenthetical has been modified to the main body.

- Page 12, Lines 211-217: When a low electron dosage (50 kGy) was applied to the CMS films, the unary gas permeabilities of relatively small gases (from He to C₂H₆) did not dramatically change (**Figure 3b**) but sharp molecular cut-off appeared between C₃ hydrocarbon gases (i.e. C₃H₆/C₃H₈ selectivity increased from 13.28 to 47.13). When the electron dosage exceeded 100kGy, the permeability of the larger gases significantly decreased (e.g., 55% reduction for CH₄ and 84% reduction for C₂H₆) compared to smaller gases—indicating the formation of tight but size-selective ultramicropores.

v) Figure 5: The loss of the ethylene permeability is due to aging effects of the polymer. The polymer is still the biggest part of your membrane after 100kGy treatment. How is it possible

that you lose much less permeability with the turbostratic graphite on top, while the selectivity goes up? It would be favourable to show the permeability loss of the ethane as well, because both data sets are used to calculate the selectivity. Also: Show the raw data and not a normalization. If it can be seen whether the CMS membrane has much lower permeability, it should become obvious why the aging is slower and why it should not be interpreted in comparison.

vi) The whole part of the aging resistance due to the CMS layer is based on wrong assumptions. The polymer is still 99% of the membrane and therefore, the aging must be the same.

It is best to respond to comment v) and vi) together; the prepared CMS membranes are totally different from polymer membrane as mentioned before, but like glassy polymers, physical aging in CMS has been observed in many studies. The pores of CMS tend to contract to reach thermodynamic equilibrium, similar to the unrelaxed free volume of glassy polymer, resulting in significant changes in transport properties during the early stages of the membrane testing.

vii) How is the reproducibility guaranteed? Please provide tables of the measured gas permeabilities for several membrane samples and include error bars to your data points.

On **Figure 3b**, we included error bars for the overall gas permeabilities of the pristine and electron-irradiated CMS membranes. Please see the preceding response to comment #iv. The error bars for small gas permeabilities (He, H₂ and CO₂) were not apparent in the overall bar graphs due to the log scale of the y-axis which includes values from 1 to 10⁴ Barrers. As a result, we include an enlarged version of He, H₂ and CO₂ permeability with error bars in Supporting Information (**Figure S9**).

Figure S9 has been added.

Supplementary Fig. 9. Enlarged version of He, H₂ and CO₂ permeability with error bars

Reviewer #2 (Remarks to the Author):

General Comments.

Oh et al. reported on the electron-irradiated CMS membranes for enhanced H₂/CO₂ and C₂H₄/C₂H₆ separation performance. Although it is well-known that the structure of carbon materials can be engineered by high-energy electron irradiation, the authors nicely demonstrated the structure-separation performance relationship of the CMS membranes. However, there are still some issues that the authors should address before publication. The detailed comments are described below.

Comment 1:

In lines 48-50 on pg 2, the authors mentioned that the limited separation performance was attributed to insufficient pore window curvature, which is not clear.

The term “insufficient pore window curvature” refers to the lack of pore confinement effect caused by the slit-like pathway of CMS. For clarity, the following parenthetical has been modified.

- Page 3, Lines 48-49: This is primarily due to insufficient pore confinement and limited populations of sub-0.5 nm pores.

Comment 2:

The authors should justify the selection of 6FDA-BPDA:DAM (1:1) for the polymer precursor. Also, specify the thickness of the films used in this work.

As described in the manuscript, CMS membranes derived from 6FDA-based polyimide have better gas and liquids separation performances than other CMS membranes made from closed and planar polyimide (e.g., Matrimid[®] and Kapton). Due to the coexistence of bulky 6FDA and planar BPDA monomers, 6FDA:BPDA(1:1)-DAM derived CMS delivers more selective pore with moderate permeability among the 6FDA-based polyimides. The pyrolyzed CMS films were 75 – 80 μm thick, whereas the precursor polymeric films were 115 μm thick due to the densification during pyrolysis process. The following parenthetical has been added to the main body.

- Page 7, Lines 113-116: Given the coexistence of bulky 6FDA and planar BPDA monomers, 6FDA:BPDA(1:1)-DAM derived CMS provides a more selective pore window with moderate permeability than other 6FDA-based polyimide such as 6FDA-DAM.^{34,35}
- Page 7, Lines 116-118: CMS membranes were fabricated with thickness of 75-80μm by pyrolyzing the dense film of 6FDA:BPDA-DAM under an ultra-high-purity (UHP) argon environment (**Figure 1a**).

The following new references were added

34. Qiu, W., Li, F. S., Fu, S. & Koros, W. J. Isomer-Tailored Carbon Molecular Sieve Membranes with High Gas Separation Performance. ChemSusChem (2020) doi:10.1002/cssc.202001567.

35. Liang, J. et al. Effects on Carbon Molecular Sieve Membrane Properties for a Precursor Polyimide with Simultaneous Flatness and Contortion in the Repeat Unit. *ChemSusChem* 13, 5531–5538 (2020).

Comment 3:

The images in Figure 3a is not very clear to understand how the electron irradiation changed the ultramicropores of CMS membranes. The authors should explain the mechanism of changing the ultramicropores of CMS membranes via electron irradiation.

Figure 1a depicts the change in ultra-microporosity by electron irradiation. The goal of **Figure 3a** is to show how the distribution of ultramicropores change before and after electron irradiation, resulting in a more concentrated pore population with a smaller size (< 0.4 nm). The red-shaded edge of the carbon strands in **Figure 3a** indicates an oxidized and compacted ultramicropore caused by electron irradiation. **Figure 3a** caption was changed and **Figure S10** was added for a better understanding of ultramicropore modification.

- Figure 1. a) Schematic description of turbostratic carbon structure with hypothetical ultramicropore size distribution change before and after electron irradiation. The red-shaded edge of the carbon strands indicates oxygen-functionalization and pore compaction.

Figure S10 has been added.

Supplementary Fig. 10. Schematic for selective gas permeation in CMS. Selective C_2H_4 permeation with blocked C_2H_6 in low irradiation dosage (left). Selective H_2/CO_2 permeation through highly irradiated CMS (right).

Comment 4:

The sp^3/sp^2 ratio of the 250 kGy-CMS membrane is still lower than that of the 100 kGy-CMS analog even at 3.75 μm in depth (Figure 3f), reflecting that the 250 kGy dosage is still strong

enough to affect the structure of CMS membranes at 3.75 μm in depth. Also, the higher electron beam intensity is expected to increase the effective thickness available for the electron irradiation as well as the sp²/sp³ ratio? If so, the authors need to provide the effective thickness for the electron irradiation as a function of the number of irradiation or the electron beam intensity.

Additional XPS depth profiles (50kGy- and 500kGy-CMS) were measured, and the oxidized layer thickness ("effective thickness") was measured using a total of 5 samples as suggested. **Supplementary Fig. 5** now includes the depth profiles of 50kGy- and 500kGy-CMS. The dashed line on the effective layer indicates a proportional relationship between effective thickness and irradiation dosage. **Supplementary Fig. 6** now shows the relationship between irradiation dose and oxidized thickness. **Figure 3f** and **Supplementary Fig. 20** now show the updated sp³/sp² ratio of the etched depth for 50kGy-CMS. The sp³/sp² ratios of the 50kGy-CMS near the surface (0s and 40s) laid between those of pristine and 100kGy-CMS, but the ratios at the membrane interior (from 80s) converged around 0.6 similar to the pristine one.

The following parenthetical has been added to the main body

- Page 10, Lines 188-189: However, 50kGy-CMS has an oxidized surface and the oxygen composition continuously decreases with the etched depth.

The following parenthetical has been modified to the main body.

- Page 10, Lines 189-193: In addition, all the 100kGy-CMS, 250kGy-CMS and 500kGy-CMS show a 3 or 4-fold increase in oxygen composition on the surface, indicating the formation of oxygen-rich carbon surfaces. When the electron irradiation increased from 50 kGy to 500 kGy, the oxygen composition and the thickness of the oxidized carbon layer also increased (from 1.61 to 4.13 μm, **Figure S6**).

Figure S5 has been updated.

Supplementary Fig. 5. X-ray photoelectron spectroscopy (XPS) depth profiling of 500°C 50kGy-, 100kGy-, and 500kGy-CMS with respect to the C, N, O composition versus sample depth.

Figure S6 has been added.

Supplementary Fig. 6. Thickness of oxidized carbon layer resulted from XPS depth profiling of 500°C pristine, 50kGy, 100kGy, 250kGy and 500kGy-CMS.

Figure 3f has been updated.

The following parenthetical has been modified to the main body.

- Page 14, Lines 271-275: At the surface, 50kGy-CMS, 100kGy-CMS and 250kGy-CMS had a low sp^3/sp^2 ratio (0.40, 0.35 and 0.28, respectively), and the ratio gradually increased to arrive at a similar value of the pristine CMS (~ 0.6) below 2.75 μm , which indicates that the penetrant molecules could undergo significant transport resistance on the electron irradiated CMS surface.

Figure S20 has been added.

Supplementary Fig. 20. XPS sp^3/sp^2 ratio of 500 °C 50kGy CMS at specific depths.

Comment 5:

The electron-irradiated CMS membrane is sort of asymmetric membrane since the effective thickness available for the electron irradiation is limited; however, the current analysis on the transport result is based on symmetric dense film structure, which is wrong. The authors should analyze the transport results by using a series model by assuming that the remaining electron-irradiation irrelevant CMS membrane would show the same separation performance as that for the pristine CMS analog, which can allow them to induce the intrinsic separation performance of the electron-irradiated CMS membranes. This comment is partly related to the previous

comment in #4 (the effective thickness for the electron irradiation as a function of the electron beam intensity).

We used the resistance in series model to analyze C₂H₄ permeability of electron irradiated CMS using TFC (thin film composite) structure, comprising of electron irradiated (“effective”) layer and pristine CMS layer.

$$R = \frac{l}{P * S}, \quad R_t = R_p + R_e$$

$$P_t = \frac{(l_p + l_e)P_p P_e}{P_p l_e + P_e l_p}$$

Here, R_t , R_p , and R_e are resistance of the total electron-irradiated CMS layer, pristine CMS layer, and effective CMS layer, respectively. Also, l_p and l_e is thickness of the pristine and effective layer. P_t , P_p and P_e are the permeability of the whole irradiated CMS (including both effective and pristine layer), pristine layer, and effective layer, respectively. The results of the estimation are shown below.

	Pristine	50kGy	100kGy	250kGy	500kGy
P_t of C ₂ H ₄	630.37	368.31	180.6	28.95	3.41
P_e of C ₂ H ₄		21.99	7.34	1.33	0.23

The P_e of the electron irradiated CMS membranes (50 – 500kGy) decreases with the irradiation dosage, revealing the effective layer contributes to the selective gas permeation. However, there are several concerns in analyzing the intrinsic separation performance of electron-irradiated CMS based on the current series model. First, as shown by the XPS depth profiling results, oxygen composition is not in a form of step function and exhibits a constantly decreasing behavior with depth, indicating that using the effective layer as an uniform resistance is not accurate. Another concern is that electron-irradiated CMS cannot be considered as pristine CMS except for the effective layer. Reviewer regarded the unoxidized layer (below the effective layer) as “irradiation irrelevant CMS”. However, the structural change of carbon material due to electron irradiation is found even in a vacuum condition (1,2), so it is not correct to consider that only the oxidized layer is affected by the electron irradiation procedure. Therefore, the transport data of the electron irradiation on CMS membranes in vacuum or oxygen-free conditions is needed for precisely analyzing the transport model-- P_p would be revised in this context. The electron accelerator in use at the moment operates in ambient conditions and requires extensive setup changes to create a vacuum. We are currently establishing an appropriate setup for oxygen-free electron irradiation for CMS membranes and the results will be shared in a follow up study.

- (1) Gaba, E. on & acti-, R. C. Encapsulated C60 in carbon nanotubes. *Nature* (1998).
- (2) Banhart, F. Formation and transformation of carbon nanoparticles under electron irradiation. *Philos. Trans. A Math. Phys. Eng. Sci.* **362**, 2205–2222 (2004).

Comment 6:

The authors should explain why the C₂H₄/C₂H₆ selectivity of the 100 kGy-CMS membrane

decreased after 36 d aging process (Figure 5b). In fact, the explanation on the enhanced aging resistance of the electron-irradiated CMS membrane is missing.

Thank you for recommending that the aging behavior of the 100kGy-CMS membrane be supplemented. C, H, N, and O composition for fresh (as prepared) and aged samples were additionally determined using elemental analysis (EA) technique to understand the chemistry of the CMS aging process (**Table S9**). The relatively higher oxygen fraction in 100kGy-CMS (9.95%) than in pristine-CMS (6.68%) is consistent with the XPS results, given that oxygen enrichment evolved mostly on the CMS surface. In this case, the CMS samples were stored in ambient air condition for aging testing for more than 30 days. The pristine CMS showed slightly increased oxygen fraction in aged samples because carbon edge sites are capable of chemisorb oxygen in ambient air condition (1,2). However, in the case of 100kGy-CMS, oxygen fraction reduced from 9.95% to 9.06 % in 30 days. This result is analogous to the loss of the oxygen-containing group in graphene oxide (GO) after aging, and can be viewed as a process in which an over-oxidized CMS surface by electron-irradiation is brought back to equilibrium structure (3).

- (1) Menendez, I. & Fuertes, A. B. Aging of carbon membranes under different environments. *Carbon* vol. 39 733–740 (2001).
- (2) Floess, J. K., Lee, K. J. & Oleksy, S. A. Kinetics of oxygen chemisorption on microporous carbons. *Energy Fuels* **5**, 133–138 (1991).
- (3) Li, C. *et al.* Effect of long-term ageing on graphene oxide: structure and thermal decomposition. *R Soc Open Sci* **8**, 202309 (2021).

Table S9 has been added.

	Weight fraction(%)			
	C	H	N	O
Fresh pristine-CMS	78.39	2.76	7.04	6.68
Aged (14 days) pristine-CMS	77.15	2.72	6.86	6.92
Fresh 100kGy-CMS	78.62	2.82	6.93	9.95
Aged (30 days) 100kGy-CMS	77.89	2.93	6.83	9.06

Supplementary Table 9. Elemental fraction of pristine- and 100kGy-CMS films pyrolyzed at 500°C

The following discussion regarding the transport mechanisms of the aged CMS samples is conceivable based on the changes in the oxygen fraction of the pristine-CMS and 100kGy-CMS before and after the aging process. The slit-like microstructure of fresh and aged electron-irradiated CMS is depicted in the newly added schematic graphic (**Figure S27**). The D_1 and D'_1 are supermicropore dimensions and D_2 , D_3 , D'_2 and D'_3 are ultramicropore of the fresh and aged electron-irradiated CMS, respectively, according to the IUPAC definition. When D_1 shrinks as a result of aging, the permeabilities of C_2H_4 and C_2H_6 in aged electron-irradiated CMS membranes decrease. During aging process, the oxygen adsorbed on the carbon edge sites degrades and the ultramicropore dimensions (D'_2 and D'_3) increase relative to the fresh sample, resulting in a decreased selectivity of the aged CMS membranes. This contrast

structural evolution of supermicropores and ultramicropores would explain the unusual transport properties of the 100kGy-CMS after aging process.

Figure S27 has been added.

Supplementary Fig. 27. Hypothetical scheme for the fresh and aged electron-irradiated CMS microstructure. The slit-like structure of CMS composed of micropore (D_1 and D'_1) and ultramicropore (D_2 , D'_2 , D_3 and D'_3).

The following parenthetical has been added to the main body.

- Page 18, Lines 356-372: Additionally, elemental analysis (EA) was performed to figure out the aging characteristic of the CMS membranes (Table S9). The aged pristine CMS (6.92%) revealed a relatively higher oxygen fraction than that of fresh pristine CMS (6.68%) because reactive carbon edges slowly chemisorb oxygen in air condition.^{48,49} The oxygen fraction in 100kGy-CMS decrease from 9.95% to 9.06% in 30 days. This result is comparable to the loss of the oxygen-containing group in aged graphene oxide (GO) and can be considered as a process where over-oxidized CMS surface rebound back to equilibrium structure.⁵⁰ The unusual aging behavior (selectivity decrease) of the electron-irradiated CMS is conceivable with hypothetical scheme based on the changes in oxygen fraction of the 100kGy-CMS after aging process (Figure S27). The D_1 and D'_1 are supermicropore dimensions and D_2 , D_3 , D'_2 and D'_3 are ultramicropore of the fresh and aged electron-irradiated CMS, respectively, according to the IUPAC definition. When aging process proceeds, the supermicropore dimension D_1 shrinks to D'_1 and entails overall permeability reduction in both C_2H_4 and C_2H_6 . Oxygen adsorbed on the carbon edge sites is less developed after aging and it makes ultramicropore dimensions (D'_2 and D'_3) increased than the fresh sample, resulting in slightly lessened selectivity of the aged CMS. That is, contrast structural evolution in supermicropore and ultramicropore explains the unique permeation behavior of 100kGy-CMS after aging process.

The following new references were added

48. Menendez, I. & Fuertes, A. B. Aging of carbon membranes under different environments. *Carbon* vol. 39 733–740 (2001).

49. Floess, J. K., Lee, K. J. & Oleksy, S. A. Kinetics of oxygen chemisorption on microporous carbons. *Energy Fuels* 5, 133–138 (1991).

50. Li, C. *et al.* Effect of long-term ageing on graphene oxide: structure and thermal decomposition. *R Soc Open Sci* **8**, 202309 (2021).

Comment 7:

The mechanical integrity is an important issue in developing carbon membranes for gas separation. The authors should provide the mechanical property of the electron-irradiated CMS membranes and compare it with that of the precursor as well as the pristine CMS membrane.

We agree with your point about the mechanical stability challenges with carbon membranes, because they are directly related to scalability and feasibility of the process. With extensive hands on experiences with the electron-irradiated CMS membranes, we believe the macroscopic mechanical properties did not change upon electron irradiation on CMS membranes. We also acknowledge that, to effectively compare mechanical properties of the precursor, pristine-CMS and electron-irradiated CMS, a suitable membrane type such as hollow fiber should be produced and examined. The hollow fiber CMS membranes are now being studied in our group using the same electron irradiation technique, and therefore the mechanical properties will be addressed in a future work.

For a similar case, a bundle of carbon fibers (not membranes, outer diameter < 10 μ m) were examined to measure the tensile property following electron irradiation (50 – 300 kGy). The tensile stress and elongation of the carbon fibers remained stable, with nearly constant values. We can deduce from this that electron irradiation of CMS membranes may have little effects on mechanical strength. The following parenthetical and references have been added to the main body.

The following parenthetical has been added to the main body.

- Page 7, Lines 120-124: From our extensive experience with electron-irradiated CMS membranes, we believe that the macro-mechanical properties were not altered upon electron irradiation of CMS membranes. Since mechanical property test on a bundle of carbon fibers (not membranes, diameter < 10 μ m) revealed stable tensile stress and elongation after electron irradiation, mechanical property of electron-irradiated CMS would be insignificantly changed.^{36,37}

The following new references were added

36. Giovedi, C., Diva Brocardo Machado, L., Augusto, M., Segura Pino, E. & Radino, P. Evaluation of the mechanical properties of carbon fiber after electron beam irradiation. *Nucl. Instrum. Methods Phys. Res. B* **236**, 526–530 (2005).

37. Park, S. K., Jung, S., Lee, D. Y., Ghim, H. & Yoo, S. H. Effects of electron-beam irradiation and radiation cross-linker on tensile properties and thermal stability of polypropylene-based carbon fiber reinforced thermoplastic. *Polym. Degrad. Stab.* **181**, 109301 (2020).

Reviewer #3 (Remarks to the Author):

The article entitled “Precision Carbon Membrane: Electron-mediated Control of Nanoporosity for Targeted Molecular Separation” presents the significance of carbon molecular sieves supported polymeric membrane for the gas separation applications. on preparation of charged Ultrafiltration membranes. It gives the detailed information about the physicochemical characterization and selectivity analysis of C₂H₄/C₂H₆ and H₂/CO₂. The article is well written, however minor revision is needed before this paper can be considered for the publication.

1. In abstract, the pronouns could be avoided and the selectivity data can be provided.

Thank you for pointing out this. The following parenthetical has been modified in the abstract.

- Page 1, Lines 15-17: High-precision tuning of the microstructure of CMS membranes is proposed by controlled electron irradiation for the separation of molecules with size differences less than 0.05 nm.
- Page 1, Lines 20-23: Fitting CMS membranes for targeted molecular separation can be accomplished by irradiation dosage control, resulting in highly-efficient C₂H₄/C₂H₆ separation for low dosages (~250kGy, with selectivity ~14) and ultra-selective H₂/CO₂ separation for high dosages (1000~2000kGy with selectivity ~80).

2. In morphology and XPS analysis (Fig.2e), there is no salient difference in pristine bean and electron beam irradiated membrane. It could be explained in detail.

SEM and TEM images maybe inaccurate to reveal significant morphological differences between pristine CMS and electron-irradiated CMS. However, unlike SEM or TEM images, the FFT patterns and GIWAXS patterns show narrowly developed CMS microstructure and provide substantial evidence of pore dimension variation after electron irradiation. Furthermore, Raman spectra determine the I_D/I_G ratio to suggest defective CMS development caused by electron irradiation, whereas XPS spectra reveal increased oxygen content at the CMS surface. Based on the reviewer’s comment, The CMS analysis was substantially enhanced in the revised manuscript as follows:

The following parenthetical has been modified to the main body

- Page 8, Lines 133-136: When the pristine CMS was exposed to 250 kGy (250kGy-CMS, hereafter) of electron irradiation, the radial intensity of the FFT patterns changed with different the number of rings, indicating the different packing degrees of the carbon strands.
- Page 9, Lines 155-158: Notably, 2D scattering image of 250kGy-CMS shows that the ‘out-of-plane’ vector (q_z) intensity in the range of 0.5 to 1.0 Å⁻¹ was mitigated, while the peak around 1.4 Å⁻¹ became strong and narrow (**Figure 2b and c**).
- Page 9, Lines 166-169: The survey scan of X-ray photoelectron spectroscopy (XPS) compares the atomic composition of the surface on the pristine CMS and 250kGy-

CMS, showing the intensified O1s peak (~530 eV) and relatively decreased C1 peak (~285eV) after electron irradiation (**Figure 2e** and **Figure S4**).

- Page 11, Lines 199-203: Therefore, the defectiveness in amorphous carbon materials (here, CMS) tends to be the opposite of the I_d/I_g ratio. Since the I_d/I_g ratio of the 250kGy-CMS ($I_d/I_g = 0.621$) is lower than that of the pristine CMS ($I_d/I_g = 0.667$), the electron irradiated CMS is assumed to be more defective overall. From this, we can conclude electron irradiation induces not only oxidization on the CMS surface but also defective carbon network within CMS.

3. The mechanism of CMS deposition and membrane formation need to discuss with literature. The thickness of deposited layer could be included and the influence of gas separation could be drafted.

CMS films were not deposited on the polymer substrate, but were created by pyrolyzing symmetric polymer films at various temperatures (500 – 675°C). Our CMS membrane is a symmetric, free-standing membrane fabricated without the use of any support elements. Furthermore, electron irradiation was applied as a post-pyrolysis treatment as a separate experiment. For a better understanding, a schematic illustration for the CMS fabrication and electron irradiation process from polymer film is included here.

Figure for Review. Schematic for CMS fabrication from symmetric polymer film and electron irradiation process on symmetric CMS film.

4. The stability of membrane before and after modification need to be discuss in detail with references. Plasticization effect of membrane with few characterizations could also be discussed.

We agree with your point about the mechanical stability challenges with carbon membranes, because they are directly related to scalability and feasibility of the process. With extensive hands on experiences with the electron-irradiated CMS membranes, we believe the macroscopic mechanical properties did not change upon electron irradiation on CMS membranes. We also acknowledge that, to effectively compare mechanical properties of the precursor, pristine-CMS and electron-irradiated CMS, a suitable membrane type such as hollow fiber should be produced and examined. The hollow fiber CMS membranes are now being studied in our group using the same electron irradiation technique, and therefore the mechanical properties will be addressed in a future work.

For a similar case, a bundle of carbon fibers (not membranes, outer diameter < 10 μ m) were examined to measure the tensile property following electron irradiation (50 – 300 kGy). The tensile stress and elongation of the carbon fibers remained stable, with nearly constant values. We can deduce from this that electron irradiation of CMS membranes may have little effects on mechanical strength. The following parenthetical and references have been added to the main body.

The following parenthetical has been added to the main body.

- Page 7, Lines 120-124: From our extensive experience with electron-irradiated CMS membranes, we believe that the macro-mechanical properties were not altered upon electron irradiation of CMS membranes. Since mechanical property test on a bundle of carbon fibers (not membranes, diameter < 10 μ m) revealed stable tensile stress and elongation after electron irradiation, mechanical property of electron-irradiated CMS would be insignificantly changed.^{36,37}

The following new references were added

36. Giovedi, C., Diva Brocardo Machado, L., Augusto, M., Segura Pino, E. & Radino, P. Evaluation of the mechanical properties of carbon fiber after electron beam irradiation. *Nucl. Instrum. Methods Phys. Res. B* **236**, 526–530 (2005).
37. Park, S. K., Jung, S., Lee, D. Y., Ghim, H. & Yoo, S. H. Effects of electron-beam irradiation and radiation cross-linker on tensile properties and thermal stability of polypropylene-based carbon fiber reinforced thermoplastic. *Polym. Degrad. Stab.* **181**, 109301 (2020).

5. Schematic of CMS on selective gas permeation can be included.

Supplementary Fig. 10 now includes a schematic for selective gas permeation through the electron-irradiated CMS.

Supplementary Fig. 10. Schematic for selective gas permeation in CMS. Selective C₂H₄ permeation with blocked C₂H₆ in low irradiation dosage (left). Selective H₂/CO₂ permeation through highly irradiated CMS (right).

6. Scope of studies and viability of gas separation could have discussed.

As stated on page 6 of the manuscript, the scope of this study includes microstructure tuning of CMS membranes using electron irradiation in order to separate gas pairs such as C₂H₄/C₂H₆ and H₂/CO₂. Additionally, binary gas permeation tests demonstrate the feasibility of electron-irradiated CMS membranes for separation of industrially relevant gas pairs (e.g., olefin/paraffin). By reflecting the comment, we have added the scope of study and viability to the discussion part.

The following parenthetical has been added to the main body.

- Page 20, Lines 381-386: The binary gas permeation test on the studied gas pairs (C₂H₄/C₂H₆ and H₂/CO₂) showed comparable separation capabilities to that of unary gas test and more realistic performances of electron-irradiated CMS membranes. H₂/CO₂ performances at high temperature (100 and 130 °C) were also calculated to confirm separation behavior and viability of the electron-irradiated CMS, indicating remarkable performance preservation in industrially relevant conditions.

REVIEWERS' COMMENTS

Reviewer #1 (Remarks to the Author):

Thanks for all your time and effort you invested in getting this paper finalized. The reviewers were all very critical and you responded very scientifically to everyone. You did everything I aksed for and answered very friendly to all my comments, explained everything in detail and added a lot of data.

Apologies for my misunderstandings, your described process is quite complex and it was not clear to me. Many of my comments were redundant because of misunderstanding. Now, everything is clear, the data is supporting the statements more than sufficient.

Your material could be of high interest for industrial applications, I hope you will transfer it from lab to pilot scale in the near future.

Accept as is.

Reviewer #2 (Remarks to the Author):

Overall, the authors' responses to my comments are satisfying and it can be published in NC.

Reviewer #3 (Remarks to the Author):

The authors addressed all the queries raised by reviewers. The quality f manuscript has been improved and the manuscript can be consider for publication.

Point-by-Point Responses

Reviewer #1 (Remarks to the Author):

Thanks for all your time and effort you invested in getting this paper finalized. The reviewers were all very critical and you responded very scientifically to everyone. You did everything I asked for and answered very friendly to all my comments, explained everything in detail and added a lot of data.

Apologies for my misunderstandings, your described process is quite complex and it was not clear to me. Many of my comments were redundant because of misunderstanding. Now, everything is clear, the data is supporting the statements more than sufficient.

Your material could be of high interest for industrial applications, I hope you will transfer it from lab to pilot scale in the near future.

Accept as is.

Your feedback was really beneficial in enhancing and clarifying our manuscript. We appreciate your time to review our manuscript.

Reviewer #2 (Remarks to the Author):

Overall, the authors' responses to my comments are satisfying and it can be published in NC.

Your recommendations and inquiries have provided us with the opportunity to add scientific depth to our work.

Reviewer #3 (Remarks to the Author):

The authors addressed all the queries raised by reviewers. The quality of manuscript has been improved and the manuscript can be considered for publication.

Your comments and suggestions helped us to revise our manuscript. We appreciate your helpful comments.